# Error-Driven Graph Augmentation for Mesh-Based PDE Surrogates

**Xuan Minh Vuong Nguyen** [1 2]  **Nissrine Akkari** [1]  **Fabien Casenave** [1]  **Jonathan Viquerat** [2]  **Elie Hachem** [2]

## Abstract

Graph Neural Networks (GNNs) on meshes have emerged as promising surrogates for computational mechanics, but standard local message passing struggles to propagate information across unstructured meshes, leading to large errors in regions with complex physics (e.g., shocks, wakes, boundary layers). Existing approaches enlarge connectivity with long-range edges chosen *a priori* via geometric heuristics or random sampling, which lack a mechanism to prioritize high-error regions and often introduce redundant communication. We propose **MiSe-GNN**, a dual-head architecture that adaptively augments graph connectivity using model-predicted *a posteriori* errors. MiSe-GNN jointly predicts physical fields and a node-wise error indicator; the predicted error is periodically converted into a hierarchy of augmentation edges via an adaptive tree that links high-error nodes to spatial pivots at multiple scales. This error-guided connectivity concentrates message passing where the surrogate is uncertain while keeping the graph sparse elsewhere, yielding a transparent and physically interpretable graph-space analogue of adaptive mesh refinement. Across industrial CFD and CSD benchmarks, MiSe-GNN consistently improves accuracy and accuracy–compute trade-offs over strong baselines, and qualitative analyses show that it routes communication toward physically challenging regions. These results establish *error-guided edge augmentation* as a robust and general design principle for long-range message passing in physics-aware GNNs.

---

[1]SAIR (Safran AI Research), Safran, Magny-Les-Hameaux, France [2]CEMEF, Mines Paris PSL, Sophia Antipolis, France. Correspondence to: Xuan Minh Vuong Nguyen <xuan-minh-vuong.nguyen@minesparis.psl.eu>.

*Proceedings of the 43$^{rd}$ International Conference on Machine Learning*, Seoul, South Korea. PMLR 306, 2026. Copyright 2026 by the author(s).

## 1. Introduction

Computational fluid dynamics (CFD) and computational structural dynamics (CSD) underpin applications ranging from aerodynamics and turbomachinery to climate modelling and biomechanics. Yet high-fidelity simulations remain extremely costly because they require fine meshes and sophisticated numerical schemes to resolve multi-scale physical phenomena (Anderson, 1995; Versteeg & Malalasekera, 2007). The prohibitive cost of repeatedly solving large nonlinear PDE systems has motivated a surge of interest in data-driven surrogates. Among these, Graph Neural Networks (GNNs) have emerged as a promising direction because they naturally operate on unstructured meshes and transport information via message passing (Gilmer et al., 2017; Zhou et al., 2021; Pfaff et al., 2021b; Sanchez-Gonzalez et al., 2020; Zhao et al., 2024). By treating mesh elements as nodes and numerical coupling as edges, GNNs offer an attractive blend of geometric flexibility and physical locality.

However, purely local message passing presents a fundamental bottleneck: nodes struggle to access distant information required to predict wakes, shocks, coherent structures, or stress concentrations. In practice, such long-range dependencies often span dozens of hops on industrial meshes, leading to well-documented issues such as oversquashing (signals from many paths being compressed) and underreaching (insufficient context from faraway regions) (Brandstetter et al., 2023; Li et al., 2020; Khemani et al., 2024; Sun et al., 2024). These failures are not merely architectural curiosities; they usually surface precisely in regions that drive engineering design and certification, where field predictions must be highly accurate. Improving long-range information flow is thus indispensable for deploying GNN surrogates in computational mechanics.

One common strategy is to enlarge the effective neighbourhood of each node by augmenting the mesh with additional long-range edges. Multiscale or hierarchical rewiring methods introduce shortcut connections so that information can traverse the domain in fewer message-passing steps (Ripken et al., 2023; Gladstone et al., 2024). For example, MGN Tree (Ripken et al., 2023) constructs a hierarchical tree over the mesh and injects multi-scale edges that strengthen global communication. Despite their success, these augmentations are usually determined from geometric heuristics and remain

fixed throughout training. They treat the domain uniformly, without asking *where* additional message-passing capacity is actually needed. Yet in real simulations, errors concentrate around shocks, discontinuities, complex boundary layers, or geometric singularities, so regions where learning and physics both become more demanding. Uniform augmentation therefore spreads a limited edge budget across both easy and hard regions, diluting capacity and yielding suboptimal accuracy.

Recent work has begun to highlight the broader role of graph structure in GNN performance. EvoMesh (Deng et al., 2025) dynamically evolves hierarchical meshes for unsteady CFD, while Adaptive Message Passing (Errica et al., 2025) and spectrum-preserving sparsification (Liang et al., 2025) adjust propagation patterns for efficiency and robustness. At a more theoretical level, Vitvitskyi et al. (Vitvitskyi et al., 2025) analyse which computational graph topologies enable expressive and efficient learning. Although these approaches reconfigure connectivity, none explicitly leverage a *learned a posteriori error indicator* to decide *where* and *how* to modify the graph during training. In this paper, we focus on mesh-based steady-state PDE surrogates.

In this work, we propose the *Multi-head Intelligent Self-edge Enhanced Graph Neural Network* (MiSe-GNN), a dual-head architecture that uses the model's own predicted error field to drive graph augmentation. Here, *self-edge* refers to edges *generated by the model itself from its predicted error*, not self-loops. MiSe-GNN periodically predicts node-level errors and injects error-guided augmentation edges that route information toward regions of high predicted error or strong physical variation (e.g., shocks, discontinuities, complex boundary layers), while avoiding unnecessary proliferation in well-predicted areas. This yields a principled, task-aware mechanism for allocating message-passing bandwidth. Unlike static hierarchical schemes that uniformly densify the domain, MiSe-GNN performs a form of neural *graph-space adaptive refinement*: the model selectively increases its own effective neighbourhood where the physics is hard, without predefined heuristics or manual supervision. Our contribution is not the use of error estimates for mesh adaptation in general, but the use of a learned pre-intervention a posteriori error signal to adapt message-passing connectivity on a fixed mesh graph, without modifying the underlying physical discretization. This distinction is important: MiSe-GNN refines the computational graph used by the neural surrogate, rather than remeshing the physical domain.

MiSe-GNN follows a two-stage cycle. During training, the error head first predicts a pre-intervention error map on the fixed base graph. Every $T$ epochs, this error map is converted by a discrete adaptive-tree routine into a fresh set of multiscale augmentation edges, and the field head is trained on the resulting augmented graph. The augmentation edges are recomputed from scratch rather than accumulated, so regions that become well predicted stop receiving extra connectivity, while the original mesh edges are always retained. At inference time, the same procedure is applied once: MiSe-GNN predicts the error on the fixed base graph, constructs the augmented graph, and then predicts the physical field on the augmented graph.

Experiments demonstrate consistent gains over state-of-the-art baselines across diverse CFD & CSD scenarios from the PLAID (Casenave et al., 2025b) collection (2D_MultiScHypEl (Staber & Casenave, 2025), 2D_profile (Casenave & Akkari, 2025), VKI-LS59 (Bucci et al., 2025), Tensile2d (Casenave et al., 2025a)), including comparisons with MeshGraph-Nets (MGN) (Pfaff et al., 2021a), MGN Tree (Ripken et al., 2023), and official PLAID benchmark models, together with ablations over augmentation frequency, error thresholds, ancestor hops, and loss weighting. Because the induced augmentations are computed deterministically from geometry and the predicted error field, they are easily visualized and offer an interpretable window into where the model allocates message-passing capacity.

**Our main contributions are:** (i) An error-guided edge augmentation algorithm that strategically strengthens connectivity in regions of high predicted error or strong physical variation, improving long-range information flow and prediction quality; (ii) A dual-head GNN and cyclic training scheme that jointly predicts field variables and local error, enabling dynamic prioritization of physically critical mesh regions; (iii) Comprehensive benchmarking on four industrial CFD & CSD datasets from PLAID, outperforming hierarchical and multiscale baselines and revealing physically interpretable patterns in capacity allocation.

Due to space constraints, full algorithmic details, additional datasets, ablations, and timing breakdowns are deferred to the Appendix.

## 2. Methodology

**Problem setup:** Each mesh sample is represented as a graph $G = (V, \mathcal{E})$ with $N$ nodes $V = \{v_i\}_{i=1}^N$, node features $\mathbf{x}_i \in \mathbb{R}^{d_x}$ where $d_x$ is the input feature dimension, and edge attributes $\mathbf{e}_{ij} \in \mathbb{R}^{d_e}$ on $(i, j) \in \mathcal{E}$ where $d_e$ is the dimension of the edge feature vector. MiSe-GNN maintains a *fixed* base graph $G_b = (V, \mathcal{E}_b)$ given by the mesh connectivity (optionally pre-enriched by a static rewiring baseline), and an *augmented* graph $G_a = (V, \mathcal{E}_b \cup \mathcal{E}^\star)$ where $\mathcal{E}^\star$ is a sparse set of error-guided multiscale augmentation edges. These edges are used as additional neural message-passing pathways; they do not alter the physical mesh or the underlying discretization. Our goal is to learn a surrogate that predicts steady-state fields $\hat{\mathbf{y}} = \{\hat{y}_i\}_{i \in V}$ while also

estimating a node-wise error indicator $\hat{\boldsymbol{\epsilon}} = \{\hat{\epsilon}_i\}_{i \in V}$, which is used to periodically update $\mathcal{E}^\star$ and improve long-range information propagation.

We also use the stacked representations $\mathbf{X}^v = [\mathbf{x}_1^\top; \ldots; \mathbf{x}_N^\top] \in \mathbb{R}^{N \times d_x}$ and for any edge set $\mathcal{E}$, $\mathbf{X}^e = [\mathbf{e}_{ij}]_{(i,j) \in \mathcal{E}} \in \mathbb{R}^{|\mathcal{E}| \times d_e}$. In particular, $(\mathbf{X}_b^v, \mathbf{X}_b^e)$ correspond to $G_b = (V, \mathcal{E}_b)$ and $(\mathbf{X}_a^v, \mathbf{X}_a^e)$ correspond to $G_a = (V, \mathcal{E}_b \cup \mathcal{E}^\star)$. Since the node set is fixed, $\mathbf{X}_b^v = \mathbf{X}_a^v = \mathbf{X}^v$.

## 2.1. Adaptive Tree Algorithm based on Error

A central hypothesis of this work is that *not all regions of the domain require equal message-passing bandwidth*. Hard-to-model regions such as shocks, wakes, contact stresses, or strong shear concentrate most of the error and demand richer communication to faithfully propagate long-range physical effects. Conversely, well-behaved regions can be serviced by the original mesh connectivity. MiSe-GNN operationalizes this idea through an error-guided adaptive binary tree that allocates augmentation edges where they provide the greatest benefit.

Predicted error should be interpreted as a diagnostic that the fixed base graph is locally insufficient, not as proof that a purely long-range shortcut is required. The adaptive tree therefore provides local, mid-range, and long-range multiscale connectivity depending on the spatial distribution of high-error nodes.

Detailed pseudocode is provided in Algorithm 1 (Appendix A.1). Figure 1 illustrates the key steps of the splitting process.

**Tree Construction Strategy:** Starting from the full domain, we recursively partition the mesh. At each step, the splitting axis is aligned with the direction of maximum geometric variance, and the split point (pivot) is selected as the node with the *highest local error* within the current partition. This choice aligns the topology of the tree with spatial and physical complexity: high-error regions are elevated higher in the hierarchy, producing multi-scale routing structures centered around physically challenging zones.

**Edge Generation:** From this hierarchy, we extract a sparse multi-level augmentation set $\mathcal{E}^\star$. At level 1, high-error nodes connect to their partition's parent pivot. At higher levels, pivots connect to their $k$-hop ancestors, creating a structured set of long-range edges that act as *communication highways*. These edges selectively increase the effective receptive field of the model where it matters, while preserving sparsity in the majority of the domain. This design differs from static hierarchical augmentations by making edge allocation *task-aware* and *error-driven*.

Importantly, a high predicted error should be interpreted as

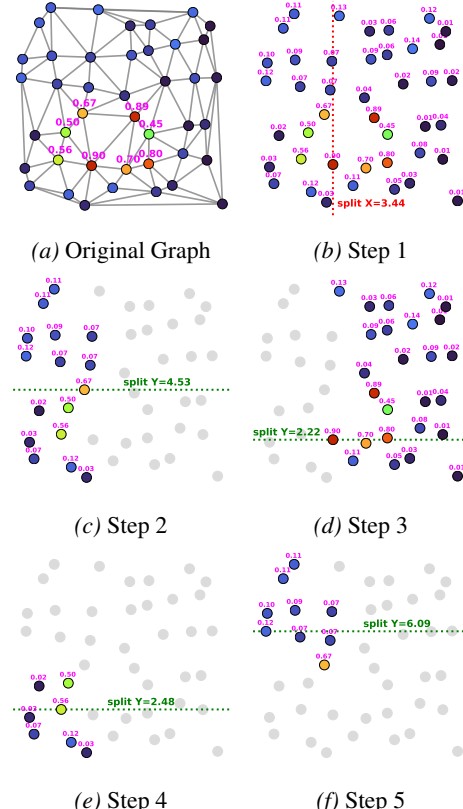

*Figure 1.* Visualization of the adaptive domain splitting process based on local error. (a) The original mesh graph with node-wise error values. (b)-(f) Sequential splits are performed, each time at the node with the highest error within the current subdomain. The splitting axis is shown as a dashed line. Node values indicate error magnitudes, with color intensity reflecting the level of error. The process continues recursively on resulting subdomains up to a specified number of levels.

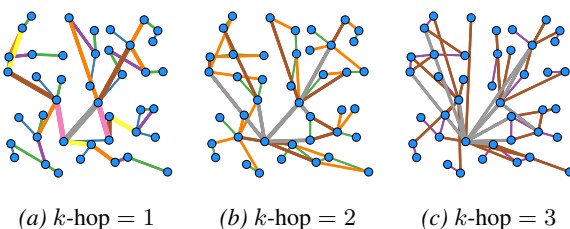

*(a) $k$-hop = 1*    *(b) $k$-hop = 2*    *(c) $k$-hop = 3*

*Figure 2.* Comparison of error-based graphs constructed with different $k$-hop values. Varying $k$ changes the graph connectivity and information propagation pattern.

a diagnostic that the fixed base graph is locally insufficient, not as proof that a purely long-range shortcut is always required. The adaptive tree therefore provides local, mid-range, and long-range multiscale connectivity depending on the spatial distribution of high-error nodes. When high-error regions are spatially concentrated, recursive partitioning places pivots more densely around those regions, naturally increasing local and mid-range connectivity; when nonlocal

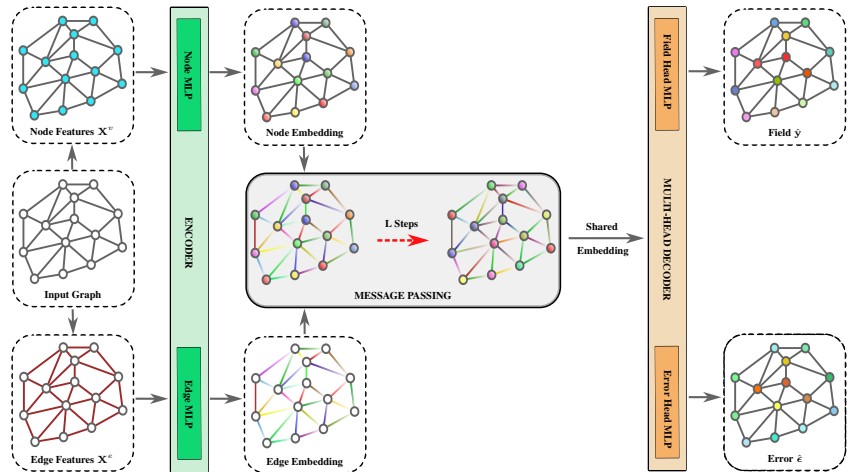

*Figure 3.* Illustration of MiSe-GNN architecture.

interactions are present, ancestor edges provide longer-range communication paths.

Figure 1 illustrates the adaptive domain splitting procedure on a randomly generated mesh graph on local error magnitudes. At each splitting step, the subdomain is partitioned along the coordinate axis exhibiting maximum variance (*split axis*), using the point with the highest local error (*pivot*). In Step 1, the domain is split along the $x$-axis at the point with error $0.9$, creating two subdomains, with the pivot point assigned to the right subdomain as defined in Algorithm 1. Subsequent splits (Steps 2–5) recursively apply this approach, each time selecting a new pivot point and split axis. Notably, in Step 5, the pivot chosen is not the highest error point $(0.67)$ but the second highest $(0.12)$, since splitting at the highest error point would result in an empty partition. Figure 2 shows the effect of the ancestor hop parameter ($k$-hop) on edge construction. Different choices of the hop parameter $k$ produce different augmentation patterns, thus enabling a controllable trade-off between long-range communication and sparsity.

Pivots are selected independently within the current subdomain. The same mesh node is not repeatedly selected within one leaf. Across descendant subdomains, a node can only be selected again if it remains the highest-error valid splitter in the smaller subdomain. If the highest-error candidate would create an empty partition, the algorithm retries with the next candidate, which explains the fallback behavior illustrated in Step 5 of Figure 1.

### 2.2. MiSe-GNN Architecture

We denote by $\mathbf{h}_i^{(l)} \in \mathbb{R}^{d_h}$ the hidden representation of node $v_i$ at message-passing layer $l$, where $d_h$ is the hidden dimension. A standard GNN updates node embeddings by aggregating messages from the neighborhood (Gilmer et al.,

2017; Battaglia et al., 2018):

$$
\mathbf{h}_i^{(l+1)} = \phi\left(\mathbf{h}_i^{(l)}, \bigoplus_{j \in \mathcal{N}(i)} \psi(\mathbf{h}_i^{(l)}, \mathbf{h}_j^{(l)}, \mathbf{e}_{ij})\right) \quad (1)
$$

While effective, this formulation is inherently local and cannot in general propagate long-range information without traversing many hops, which is insufficient for CFD phenomena involving non-local interactions (e.g., coherent vortices, turbulence, shear layers, structural stresses) (Brandstetter et al., 2023; Li et al., 2020; Kochkov et al., 2021).

To address this, MiSe-GNN introduces a shared *encoder–processor* with two task-specific *decoders* (Figure 3). The encoder maps $\mathbf{X}^v$ and $\mathbf{X}^e$ to latent representations via MLPs ($\phi_v, \phi_e$). A processor $\Psi$ composed of multiple message-passing layers captures physical interactions. The latent representation then branches into a field head $h_{\text{fld}}$, producing $\hat{\mathbf{y}}$, and an error head $h_{\text{err}}$, producing $\hat{\boldsymbol{\epsilon}}$. The latter serves as a learned a posteriori estimator guiding graph augmentation, enabling the model to *self-diagnose* and *self-refine* its connectivity.

### 2.3. Cyclic Training and Inference

MiSe-GNN separates error diagnosis from field prediction. The error head is always supervised on the fixed base graph $G_b$, so that it learns where the original message-passing graph fails before intervention. The field head is trained on the augmented graph $G_a = (V, \mathcal{E}_b \cup \mathcal{E}^\star)$, where $\mathcal{E}^\star$ is periodically rebuilt from the current predicted error map.

Within each $T$-epoch window, we keep the augmented edge set $\mathcal{E}^\star$ fixed and compute:

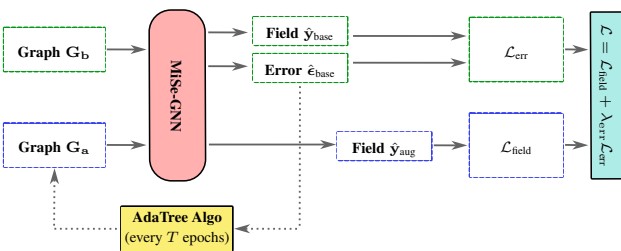

*Figure 4.* Cyclic training.

$$\hat{\boldsymbol{\epsilon}}_{\text{base}} = h_{\text{err}}\Big(\Psi\big(\phi_v(\mathbf{X}_b^v), \phi_e(\mathbf{X}_b^e)\big)\Big) \tag{2}$$

$$\hat{\mathbf{y}}_{\text{base}} = h_{\text{fld}}\Big(\Psi\big(\phi_v(\mathbf{X}_b^v), \phi_e(\mathbf{X}_b^e)\big)\Big) \tag{3}$$

$$\hat{\mathbf{y}}_{\text{aug}} = h_{\text{fld}}\Big(\Psi\big(\phi_v(\mathbf{X}_a^v), \phi_e(\mathbf{X}_a^e)\big)\Big) \tag{4}$$

The base graph prediction $\hat{\mathbf{y}}_{\text{base}}$ is used only to construct the error target. Let $\text{sg}(\cdot)$ denote the stop gradient operator. We define the scalar node-wise error target as:

$$\boldsymbol{\epsilon}^\star = |\text{sg}(\hat{\mathbf{y}}_{\text{base}}) - \mathbf{y}| \,/\, \max(\mathbf{y}) \tag{5}$$

**Remark:** The error prediction is always computed on $G_b$, making the rewiring signal stable and independent of changes in $\mathcal{E}^\star$. At the end of each cycle, we rebuild $\mathcal{E}^\star$ using $\hat{\boldsymbol{\epsilon}}_{\text{base}}$ (Algorithm 1) and update the augmented graph.

**Decoupled supervision:** We evaluate the field loss on $G_a$ and the error loss on $G_b$. The total loss is:

$$\mathcal{L} = \underbrace{\text{MSE}\big(\hat{\mathbf{y}}_{\text{aug}}, \mathbf{y}\big)}_{\mathcal{L}_{\text{field-aug}}} + \lambda_{\text{err}} \underbrace{\text{MSE}\big(\hat{\boldsymbol{\epsilon}}_{\text{base}}, \boldsymbol{\epsilon}^\star\big)}_{\mathcal{L}_{\text{err-base}}} \tag{6}$$

where $\lambda_{\text{err}}$ is a tunable hyperparameter controlling the weight of the error supervision term. By default, we set $\lambda_{\text{err}} = 1$, but in practice its value should be calibrated to balance the two objectives. The effect and selection of $\lambda_{\text{err}}$ is further analyzed in the Ablation Study (Appendix E.2).

Gradients from $\mathcal{L}_{\text{err-base}}$ flow through $\hat{\boldsymbol{\epsilon}}_{\text{base}}$, the error head, and the shared encoder-processor, but not through $\boldsymbol{\epsilon}^\star$, $\hat{\mathbf{y}}_{\text{base}}$, or the field head. The adaptive-tree construction is discrete and is not differentiated through.

At each augmentation update, $\mathcal{E}^\star$ is initialized as an empty set and rebuilt from the current predicted error map $\hat{\boldsymbol{\epsilon}}_{\text{base}}$. Thus augmentation edges are refreshed rather than accumulated: nodes that no longer exceed the error threshold lose their previously assigned augmentation edges. The original mesh edges $\mathcal{E}_b$ are always retained because they encode the local physical discretization couplings. By default, we use $T = 10$, which provides a practical balance between responsiveness to the changing error map and the cost of rebuilding the augmented graph.

This design stabilizes error prediction on $G_b$ and lets the field head leverage the richer connectivity of $G_a$.

During inference (Figure 7), given an unseen graph $G_b$, MiSe-GNN first predicts $\hat{\boldsymbol{\epsilon}}_{base}$, constructs the augmented graph via $\text{AdaptiveTreeAugment}(G_b, \hat{\boldsymbol{\epsilon}}_{base})$, and finally predicts $\hat{\mathbf{y}}_{aug}$ on $G_a$. This yields a self-adaptive surrogate that allocates communication capacity where it is most needed. We quantitatively verify that $\hat{\boldsymbol{\epsilon}}_{base}$ reliably ranks and localizes true errors (e.g., Spearman $r_s$ and Top-5% error-mass capture $M_{5\%}$; definitions in Appendix G).

## 3. Experiments and Results

### 3.1. Setup

**Baselines:** We compare MiSe-GNN against strong and widely used GNN surrogates as well as competitive non-graph baselines. **MeshGraphNets (MGN)** (Pfaff et al., 2021a) is a standard message-passing GNN tuned for PDE surrogates and has become a de facto reference architecture for mesh-based simulation. **MGN Tree** (Ripken et al., 2023) augments MGN with a hierarchical rewiring scheme that introduces multi-scale connections to accelerate long-range propagation and directly targets the oversquashing/under-reaching regime. On the PLAID benchmarks we additionally report public leaderboard models, including the neural-field based **MARIO** (Catalani et al., 2025), the Gaussian-process based **MMGP** (Casenave et al., 2024), the vision-inspired **Vi-Transformer** (Dosovitskiy et al., 2021), and the operator-learning baseline **FNO** (Li et al., 2021). Together, these systems span message-passing GNNs, neural operators, and neural fields, providing a diverse and competitive testbed. Configuration details are given in Appendix B.2.

**Metric:** Following the PLAID (Casenave et al., 2025b) benchmark, we report the Relative Root Mean Square Error (RRMSE) for both mesh fields and scalar quantities. This metric normalizes errors by the magnitude of the reference solution and by the number of nodes, enabling fair comparison across samples with different mesh resolutions and scales. For mesh fields, we use

$$\text{RRMSE}_f(\mathbf{f}_{\text{ref}}, \mathbf{f}_{\text{pred}}) = \left( \frac{1}{n_\star} \sum_{i=1}^{n_\star} \frac{\|\mathbf{f}_{\text{ref}}^i - \mathbf{f}_{\text{pred}}^i\|_2^2}{N^i \|\mathbf{f}_{\text{ref}}^i\|_\infty^2} \right)^{1/2} \tag{7}$$

Here $N^i$ is the number of mesh nodes in sample $i$, $n_\star$ is the number of test samples, and $\|\mathbf{f}_{\text{ref}}^i\|_\infty$ denotes the maximum absolute component of $\mathbf{f}_{\text{ref}}^i$. For scalar outputs (e.g., integrated forces or effective energies), we use the analogous

relative RMSE:

$$\text{RRMSE}_s(\mathbf{s}_{\text{ref}}, \mathbf{s}_{\text{pred}}) = \left( \frac{1}{n_\star} \sum_{i=1}^{n_\star} \frac{|s_{\text{ref}}^i - s_{\text{pred}}^i|^2}{|s_{\text{ref}}^i|^2} \right)^{1/2}$$
(8)

All results reported in Section 3.2 use these PLAID-standard metrics.

**Datasets:** We evaluate on four PLAID subsets spanning both CFD and CSD regimes, using the official train/test splits and observable definitions. These datasets cover 2D hyperelastic homogenization, external aerodynamics, turbomachinery flows, and nonlinear structural mechanics, with varying mesh resolutions and topologies. A brief description is provided in Appendix B.1. For each dataset and each target observable (field or scalar), we train a separate MiSe-GNN model and baseline model, ensuring a one-to-one comparison at the task level.

### 3.2. Results

**Experimental protocol:** We follow the PLAID benchmark (Casenave et al., 2025b) (official train/test splits, observable definitions) and report RRMSE (lower is better) as defined in Eqs. (7)–(8). Unless otherwise noted, all graph-based baselines use *10* message-passing steps (MP) and hidden size *32*. For fair comparison, MiSe-GNN and all GNN baselines share *identical* input node and edge features, so that any differences in performance can be attributed to the architecture and connectivity, rather than feature engineering.

We then systematically study two axes of model capacity: (i) scaling the depth/width to *15* MP / hidden size *80*, and (ii) enriching inputs with geometric encodings. For the latter, we consider *Directional Integrated Distance (DID)* (Jessica et al., 2023), *Sinusoidal Basis (Sinus)*, and *Spherical Harmonics (Sph)* (Helwig et al., 2024), using the precise definitions given in Appendix C.2. This allows us to disentangle the gains due to MiSe-GNN's error-guided connectivity from those due to stronger input representations.

For datasets with exploitable geometric structure, we additionally evaluate deterministic mesh-derived features (**OurFeat**). For 2D_MultiScHypEl, **OurFeat** encodes hole-aware geometry (e.g., polar coordinates to nearby holes, local porosity/gap indicators, and hole-occlusion edge flags; see Appendix C.3.1), and we design analogous specimen-aware features for Tensile2d (Appendix C.3.2). In all **OurFeat** settings, we keep the architecture and training protocol fixed and *only* augment the input feature channels.

We report detailed results on 2D_MultiScHypEl and 2D_profile in the main paper, and provide complete results on VKI-LS59 and Tensile2d in Appendix D.

Across all four datasets, **MiSe-GNN** consistently improves accuracy over **MGN** and **MGN Tree** under the same message-passing budget.

#### 3.2.1. 2D_MULTISCHYPEL DATASET

**Base setting (10 MP / hidden size 32):** Table 1 reports per-field RRMSE for the base configuration shared across MiSe-GNN and both MGN baselines. Averaging the per-field relative reduction, **MiSe-GNN** reduces RRMSE by **35.1%** vs. **MGN** and by **20.6%** vs. **MGN Tree**, demonstrating that error-guided augmentation substantially improves accuracy at fixed message-passing budget. The largest improvements are obtained on the displacement components {u1,u2} (up to $\sim 56\%$), which are known to require long-range information propagation (transport of boundary interaction across the porous structure). Gains on psi are smaller but consistent ($\sim 21\%$), consistent with its smoother spatial profile and weaker long-range couplings.

*Table 1.* Results on 2D_MultiScHypEl (10 MP / hidden size 32).

| Model | u1 | u2 | P11 | P12 | P22 | P21 | psi |
|---|---|---|---|---|---|---|---|
| MGN | 0.0419 | 0.0435 | 0.0365 | 0.0670 | 0.0368 | 0.0679 | 0.0455 |
| MGN Tree | 0.0249 | 0.0238 | 0.0315 | 0.0670 | 0.0326 | 0.0634 | 0.0405 |
| **MiSe-GNN** | **0.0185** | **0.0190** | **0.0258** | **0.0503** | **0.0260** | **0.0482** | **0.0360** |
| $\Delta_{\text{MGN}}$ | 55.85% | 56.32% | 29.32% | 24.93% | 29.35% | 29.01% | 20.88% |
| $\Delta_{\text{MGN Tree}}$ | 25.70% | 20.17% | 18.10% | 24.93% | 20.25% | 23.97% | 11.11% |

Figure 5 shows qualitative comparisons for u1. MiSe-GNN reduces high-error regions around hole boundaries, confirming that routing capacity toward hard regions improves surrogate fidelity.

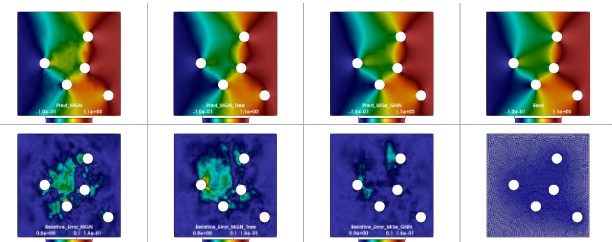

*Figure 5.* Qualitative comparison on 2D_MultiScHypEl. Top row: predicted fields, bottom row: relative errors. From left to right: **MGN**, **MGN Tree**, **MiSe-GNN**, and ground truth.

**Capacity scaling & feature engineering (15 MP / hidden size 80):** To assess robustness under increased budget, we scale capacity to (15/80) and enrich the node/edge features. Table 2 compares generic geometric encodings (*DID / Sinus / Sph*) and dataset-specific deterministic features (**OurFeat**). Among generic encodings, *DID* yields the strongest improvements, particularly on stress-related fields {P11,P12,P22,P21}, lowering the macro-average RRMSE to 0.0251 ($-21.5\%$ vs. MiSe-GNN 10/32). **OurFeat** alone provides comparable macro-average reduc-

tion and improves both displacement and stress components.

*Table 2.* Results on `2D_MultiScHypEl` with capacity scaling and feature engineering (15 MP / hidden size 80).

| Model | u1 | u2 | P11 | P12 | P22 | P21 | psi |
|---|---|---|---|---|---|---|---|
| MiSe-GNN | 0.01464 | 0.01503 | 0.02174 | 0.04812 | 0.02181 | 0.04725 | 0.03397 |
| MiSe-GNN (DID) | 0.01494 | 0.01673 | 0.01928 | 0.03620 | 0.01878 | 0.03524 | 0.03449 |
| MiSe-GNN (DID+Sinus+Sph) | 0.01555 | 0.01668 | 0.02167 | 0.03654 | 0.02321 | 0.03713 | 0.03394 |
| MiSe-GNN (OurFeat) | 0.01248 | 0.01445 | 0.01851 | 0.03814 | 0.01864 | 0.03883 | 0.03473 |
| **MiSe-GNN (OurFeat+DID)** | **0.01203** | 0.01450 | **0.01730** | **0.03119** | **0.01764** | **0.03140** | **0.03282** |
| MiSe-GNN (OurFeat+DID+Sinus) | 0.01274 | **0.01403** | 0.01763 | 0.03184 | 0.01781 | 0.03198 | 0.03333 |
| MiSe-GNN (OurFeat+DID+Sinus+Sph) | 0.01260 | 0.01465 | 0.01780 | 0.03188 | 0.01773 | 0.03208 | 0.03347 |
| MGN (OurFeat+DID) | 0.01638 | 0.01761 | 0.02258 | 0.03779 | 0.02318 | 0.03635 | 0.03420 |
| MGN Tree (OurFeat+DID) | 0.01375 | 0.01597 | 0.02165 | 0.03605 | 0.02206 | 0.03881 | 0.03436 |

Importantly, **OurFeat + *DID*** produces the best overall configuration: it achieves the lowest macro-average RRMSE (0.0224, a 29.9% reduction vs. MiSe-GNN 10/32) and attains the best results on 6/7 fields {u1,P11,P12,P22,P21,psi}, while **OurFeat + *DID* + *Sinus*** achieves the best {u2}. These results suggest that **OurFeat** and *DID* are complementary: **OurFeat** injects domain structure (hole-aware geometry) whereas *DID* provides directional boundary awareness. Together, they enhance the model's ability to resolve both global and local physical interactions.

To verify that the gains are not merely due to stronger geometric inputs, we additionally run **MGN** and **MGN Tree** with the same strongest feature set (**OurFeat + *DID***) at the same message-passing budget (15 MP / hidden size 80). Even under this enhanced setting, **MiSe-GNN** remains clearly superior, improving the macro-average RRMSE from 0.0269 (**MGN**) and 0.0261 (**MGN Tree**) down to 0.0224. This indicates that the accuracy gains primarily stem from error-guided connectivity augmentation (i.e., allocating long-range communication to persistently hard regions), rather than from feature engineering alone. In particular, the largest margins persist on stress-related quantities, supporting the interpretation that error-guided edges help propagate information across the porous microstructure where local geometric complexity induces long-range couplings.

**Comparison to PLAID benchmarks:** Table 3 situates MiSe-GNN within the PLAID leaderboard. MiSe-GNN attains the best *Total Error* (**0.0209**) and achieves the best result on {P11,P12,P22,P21} (stress-related quantities) and `effective energy`, while AUGUR leads on {u1,u2,psi}. This specialization is consistent with the architectural designs: MiSe-GNN allocates additional message-passing bandwidth toward regions of high predicted error, which in this dataset correspond to stress concentrations around the porous microstructure, while AUGUR excels on smoother displacement fields. Overall, MiSe-GNN achieves the top ranking with balanced performance and the strongest results on physically hard fields.

*Table 3.* `2D_MultiScHypEl` PLAID benchmarks. (*) MiSe-GNN (15/80) results represent the best performance across all tested feature engineering configurations. **Bold** indicates best, underline indicates second best. Results obtained on 14/10/2025.

| Rank | ID | Total Error | u1 | u2 | P11 | P12 | P22 | P21 | psi | effective energy |
|---|---|---|---|---|---|---|---|---|---|---|
| 1 | MiSe-GNN* | **0.0209** | 0.0120 | 0.0140 | **0.0173** | **0.0312** | **0.0176** | **0.0314** | 0.0328 | **0.0111** |
| 2 | Augur | 0.0221 | **0.0109** | **0.0114** | 0.0208 | 0.0336 | 0.0212 | 0.0330 | **0.0274** | 0.0188 |
| 3 | FNO | 0.0302 | 0.0115 | 0.0117 | 0.0353 | 0.0513 | 0.0359 | 0.0510 | 0.0329 | 0.0120 |
| 4 | Vi-Transformer | 0.0325 | 0.0173 | 0.0172 | 0.0337 | 0.0581 | 0.0343 | 0.0571 | 0.0312 | 0.0113 |
| 5 | Unet | 0.0350 | 0.0291 | 0.0283 | 0.0349 | 0.0500 | 0.0498 | 0.0347 | 0.0350 | 0.0180 |
| 6 | MARIO | 0.0573 | 0.0336 | 0.0377 | 0.0536 | 0.1067 | 0.0539 | 0.1053 | 0.0456 | 0.0220 |

### 3.2.2. `2D_PROFILE` DATASET

**Base setting (10 MP / hidden size 32):** Table 4 summarizes the base configuration results. **MiSe-GNN** improves over **MGN** by **25.94%–53.53%** (avg. **37.7%**) and over **MGN Tree** by **21.74%–32.60%** (avg. **25.7%**) across four fields. The largest margins occur on `Pressure` vs. MGN ($-53.5\%$) and on `Mach` vs. MGN Tree ($-32.6\%$). These gains are consistent with the trends observed on `2D_MultiScHypEl`: `Pressure` and `Mach` exhibit strong non-local dependencies due to shock-induced compression and boundary-induced acceleration around the airfoil, which benefit from error-driven augmentation. Figure 6 illustrates qualitative differences on `Pressure`, where MiSe-GNN reduces error in the shock region and downstream wake.

*Table 4.* Results on `2D_profile` (10 MP / hidden size 32).

| Model | Mach | Pressure | Velocity-x | Velocity-y |
|---|---|---|---|---|
| MGN | 0.0766 | 0.0624 | 0.0817 | 0.0559 |
| MGN Tree | 0.0678 | 0.0385 | 0.0739 | 0.0529 |
| **MiSe-GNN** | **0.0457** | **0.0290** | **0.0564** | **0.0414** |
| $\Delta_{\text{MGN}}$ | 40.34% | 53.53% | 30.97% | 25.94% |
| $\Delta_{\text{MGN Tree}}$ | 32.60% | 24.68% | 23.68% | 21.74% |

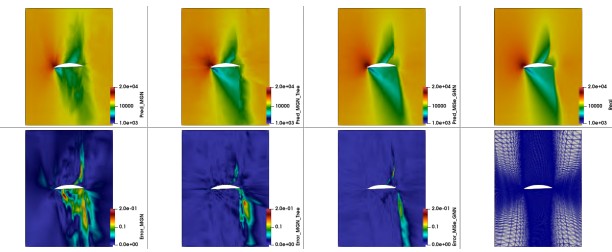

*Figure 6.* Qualitative comparison on `2D_profile`. Top row: predicted fields, bottom row: relative errors. From left to right: **MGN**, **MGN Tree**, **MiSe-GNN**, and ground truth.

**Capacity scaling & feature engineering (15 MP / hidden size 80):** Scaling to (15/80) reduces the macro-average RRMSE from 0.0431 (MiSe-GNN 10/32) to **0.0372** ($-\textbf{13.71\%}$). Adding *DID* features further lowers the macro-average to **0.0315** ($-\textbf{27.1\%}$ vs. MiSe-GNN 10/32), as shown in Table 5. Combining *DID+Sinus+Sph* yields the best macro-average overall, **0.0295** ($-\textbf{31.5\%}$), winning on `Mach` and both velocity components, while *DID* remains best on `Pressure`. These patterns indicate that directional

and harmonic encodings inject useful geometric bias for aerodynamic flows, complementing MiSe-GNN's connectivity refinements.

*Table 5.* Results on `2D_profile` with capacity scaling and feature engineering (15 MP / hidden size 80).

| Model | Mach | Pressure | Velocity-x | Velocity-y |
|---|---|---|---|---|
| MiSe-GNN | 0.04120 | 0.02242 | 0.04914 | 0.03609 |
| MiSe-GNN (DID) | 0.03447 | **0.02123** | 0.03789 | 0.03222 |
| **MiSe-GNN (DID+Sinus+Sph)** | **0.03088** | 0.02543 | **0.03576** | **0.02603** |
| MGN (DID+Sinus+Sph) | 0.03869 | 0.02746 | 0.04600 | 0.03018 |
| MGN Tree (DID+Sinus+Sph) | 0.04147 | 0.02838 | 0.04709 | 0.03328 |

We further strengthen the comparison by equipping **MGN** and **MGN Tree** with the same *DID+Sinus+Sph* encodings (Table 5). Even with identical input representations, **MiSe-GNN** retains a sizable advantage: the macro-average RRMSE is 0.0295 versus 0.0356 (**MGN**) and 0.0376 (**MGN Tree**), corresponding to relative reductions of 17.0% and 21.4%, respectively. This supports the main claim that the improvements are driven by error-guided connectivity (routing message passing to shocks/wakes), not by feature engineering alone.

Benchmarking against these strengthened baselines reveals that geometric hierarchies are not a silver bullet: **MGN Tree** even underperforms **MGN** on `2D_profile` (0.0376 vs. 0.0356). This indicates that indiscriminate long-range edges can introduce redundancy, highlighting the necessity of **MiSe-GNN**'s *selective* strategy, which routes information only where the error dynamics demand it.

**Comparison to PLAID benchmarks:** Table 6 positions MiSe-GNN within the PLAID leaderboard. MiSe-GNN achieves the best *Total Error* (**0.0285**), leading on `Mach`, `Velocity-x`, and `Velocity-y`, while MARIO leads on `Pressure`. Notably, `Pressure` is strongly tied to enforcing thermodynamic closure and is sensitive to global consistency, which explains why latent neural field approaches such as MARIO perform well on this particular observable. Overall, MiSe-GNN attains the top ranking with balanced performance across all four quantities.

*Table 6.* `2D_profile` PLAID benchmarks. (*) MiSe-GNN (15/80) results represent the best performance across all tested feature engineering configurations. **Bold** indicates best, underline indicates second best. Results obtained on 14/10/2025.

| Rank | ID | Total Error | Mach | Pressure | Velocity-x | Velocity-y |
|---|---|---|---|---|---|---|
| 1 | **MiSe-GNN**[*] | **0.0285** | **0.0309** | 0.0212 | **0.0358** | **0.0260** |
| 2 | MARIO | 0.0307 | 0.0337 | **0.0156** | 0.0379 | 0.0355 |
| 3 | Vi-Transformer | 0.0309 | 0.0360 | 0.0167 | 0.0403 | 0.0307 |
| 4 | MMGP | 0.0365 | 0.0439 | 0.0208 | 0.0471 | 0.0342 |
| 5 | Augur | 0.0425 | 0.0469 | 0.0248 | 0.0538 | 0.0445 |
| 6 | FNO | 0.0972 | 0.0988 | 0.0785 | 0.1148 | 0.0967 |

### 3.2.3. ERROR-GUIDED INFERENCE BEHAVIOR

To better understand how MiSe-GNN uses its internal error predictions at test time, we visualize a representative test sample from the `2D_profile` dataset in Figure 7. The top row shows the ground truth field $y$ together with the relative per-node error obtained on the base graph (*Real Error*) and after one round of error-guided augmentation (*Aug Error*). The bottom row depicts the corresponding message-passing graphs: on the left, the original mesh graph, and on the right, the augmented graph constructed from the predicted error field $\hat{\epsilon}_{\text{base}}$ (cf. Section 2.3).

We observe that the learned error head acts as a physics-aware feature detector: it successfully highlights regions of strong gradients, including the shock and downstream wake, while assigning low scores to laminar free-stream areas. The adaptive tree then concentrates augmentation edges $\mathcal{E}^\star$ almost exclusively in these difficult regions. This supports our core hypothesis that MiSe-GNN can *request additional communication bandwidth where the flow is hardest to model*, rather than distributing it uniformly. Quantitatively, the error head shows strong fidelity on `2D_profile`, with mean Spearman correlation 0.619 and Top-5% error-mass capture 0.753. This confirms that the model prioritizes dominant error regions even under imperfect ranking. Crucially, this reliability holds across all four benchmarks (Appendix G) with an average mass capture 0.666, validating $\hat{\epsilon}_{base}$ as a consistent driver for adaptive graph augmentation.

After augmenting the graph and recomputing the prediction, both the magnitude and spatial support of the high-error zone shrink markedly (compare *Real Error* and *Aug Error* in Figure 7). This qualitative behavior indicates that MiSe-GNN can (i) diagnose where its own predictions are unreliable and (ii) selectively allocate message-passing capacity to those regions, yielding more accurate and physically consistent predictions. In this sense, the model implements a form of self-refinement or *error-driven graph AMR*, analogous to adaptive mesh refinement in CFD but acting directly in graph space.

### 3.3. Analysis and Discussion

The preceding inference analysis shows that the learned error head provides a physics-aware signal for graph augmentation (Figure 7, Figure 29). Here we summarize the broader empirical implications of this mechanism.

**Robustness across regimes:** MiSe-GNN generalizes across both fluid and solid mechanics. On `VKI-LS59`, it reduces the challenging turbulent-viscosity field `nut` by 60.6% compared with MGN (Table 14), while on `Tensile2d` it reduces the `U1` displacement error by 94.6% compared with MGN (Table 17). These results suggest that error-guided augmentation addresses a domain-general bottleneck of local message passing: insufficient communication capacity in physically difficult regions.

**Efficiency:** Although MiSe-GNN increases per-epoch cost

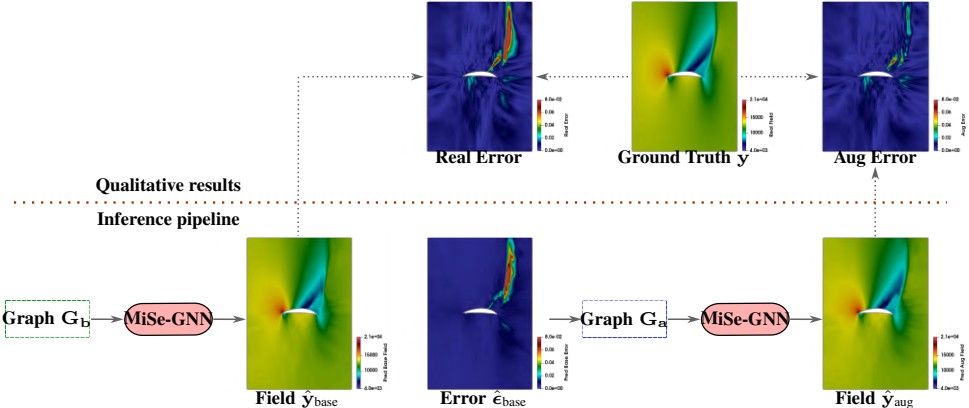

*Figure 7.* Error-guided inference on a representative `2D_profile` test sample (`Pressure` field).

(dual-head training and extra messages), the budget study shows better accuracy–time trade-offs and faster time-to-target on large-scale datasets like `2D_profile` (Appendix E.1). MiSe-GNN reaches target error thresholds up to $3\times$ faster than baselines (Table 21). By resolving high-error regions early, it amortizes auxiliary overheads and offers superior accuracy-time trade-offs.

**Connection to low-homophily regions:** Sharp gradients, shocks, wakes, discontinuities, boundary layers, and stress concentrations can be viewed as locally low-homophily regimes: neighboring mesh nodes may have strongly dissimilar target values despite being adjacent in the physical discretization. Purely local aggregation can therefore mix incompatible information in such regions. Standard local message passing can struggle in such regions because purely local aggregation mixes information across rapidly changing fields. MiSe-GNN mitigates this issue by using the predicted pre-intervention error map to allocate sparse multiscale augmentation edges where the base graph is likely to be unreliable. A more formal homophily/heterophily analysis for mesh-based PDE surrogates is left for future work.

**Equal-budget augmentation controls:** The gains are not due to adding more edges alone. Under matched average augmentation edge budgets, MiSe-GNN outperforms random, distance-based, and static hierarchical alternatives across all datasets. Table 7 summarizes the comparison against the best non-error-guided control, while Appendix E.4 provides the full per-dataset, per-field results and qualitative edge visualizations.

## 4. Conclusions and Limitations

We introduced **MiSe-GNN**, a dual-head mesh GNN that predicts PDE solution fields and node-wise *a posteriori* errors, and periodically converts the predicted errors into sparse multiscale augmentation edges. By routing long-

*Table 7.* MiSe-GNN compared with the best non-error-guided equal-budget edge control. Lower is better (Macro RRMSE).

| Dataset | Best Control | MiSe-GNN | Gain |
|---|---|---|---|
| `2D_profile` | 0.0368 | **0.0295** | 19.7% |
| `2D_profile Uniform` | 0.0250 | **0.0207** | 17.3% |
| `2D_MultiScHypEl` | 0.0242 | **0.0224** | 7.4% |
| `VKI-LS59` | 0.0191 | **0.0156** | 18.3% |
| `Tensile2d` | 0.0039 | **0.0031** | 21.6% |

range message-passing capacity toward physically challenging regions, MiSe-GNN improves accuracy and accuracy–compute trade-offs over strong baselines on industrial CFD and CSD benchmarks.

**Limitations and Outlook:** MiSe-GNN introduces additional overhead from edge construction, increased message passing, and the second prediction head. Its benefit also depends on the calibration of the error predictor; under distribution shift, miscalibrated error maps may lead to sub-optimal graph augmentation. The added edges should be interpreted as learned information pathways rather than a physics-exact discretization of characteristic, streamline, or stress-transport directions. The current topology is undirected, and anisotropy is handled through edge attributes such as relative displacement and geometric encodings. Physics-aware directional rewiring is therefore a natural future extension. In addition, the current implementation performs two forward passes at inference time, one on the base graph for error prediction and one on the augmented graph for field prediction. Reusing intermediate representations from the first pass could reduce this cost. Finally, extending MiSe-GNN to very large 3D meshes will require more scalable dynamic connectivity and tree construction, quantifying run-to-run variability with many random seeds remains computationally expensive on large industrial meshes, and a promising direction is hybrid AMR that combines refinement in physical space with connectivity refinement in graph space.

## Impact Statement

This work introduces a mechanism for error-driven graph augmentation that improves long-range message passing in mesh-based PDE surrogates. The proposed approach has the potential to reduce reliance on costly numerical solvers in engineering pipelines by enabling accurate and interpretable surrogate models with better accuracy–compute trade-offs. Beyond computational mechanics, the method contributes to graph learning by showing how model feedback can be used to adapt connectivity, suggesting a pathway toward more reliable machine learning systems that diagnose and correct their own failure modes. Potential risks are limited to the misuse of surrogate predictions in high-stakes engineering scenarios without proper verification; such risks can be mitigated through conservative deployment, uncertainty monitoring, and domain expert oversight.

**Code availability:** A release landing page for this work is available at https://github.com/XuanMinhVuongNGUYEN/mise_gnn. The full PyTorch implementation, training scripts, and configuration files are pending institutional open source clearance and will be released through this page.

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

# A. Methodological Details

## A.1. Adaptive Tree Algorithm

---

**Algorithm 1** Adaptive Tree Construction based on Error

---

**Input:** Mesh point $\mathbf{P} = \{p_i\}_{i=1}^N$, errors $\boldsymbol{\epsilon} = \{\epsilon_i\}_{i=1}^N$, max tree depth $n_{\text{levels}}$, min leaf size $n_{\min}$, error threshold $\tau_{\text{err}}$, ancestor hops $k$

**Output:** Edge set $\mathcal{E}$ grouped by level

**Helper operations:** Leaf($\mathbf{P}, \boldsymbol{\epsilon}$): Creates leaf storing a subset of mesh vertices $\mathbf{P}$ (with **global mesh indices**) and errors $\boldsymbol{\epsilon}$
                     Node($\mathcal{L}_{\text{left}}, \mathcal{L}_{\text{right}}$, pivot, axis): Creates internal node storing references to child leaves $\mathcal{L}_{\text{left}}, \mathcal{L}_{\text{right}}$, pivot index, and split axis
                     AncestorClamp($n, k$): returns $k$-hop ancestor of **tree node** $n$; if fewer than $k$ hops exist, return the root **tree node**

Initialize root $\mathcal{T} \leftarrow$ Leaf($\mathbf{P}, \boldsymbol{\epsilon}$)

*// Phase 1: Top-down Tree Construction*

**for** $\ell = 1$ **to** $n_{\text{levels}}$ **do**
  **for each** leaf $\mathcal{L} \in \mathcal{T}$ **do**
    Let $P_{\mathcal{L}}$ and $\epsilon_{\mathcal{L}}$ denote the points and errors in leaf $\mathcal{L}$
    **if** $|\mathcal{L}| \geq n_{\min}$ **and** $\max(\epsilon_{\mathcal{L}}) > 0$ **then**
      Select split axis with max variance: $a^* \leftarrow \arg\max_a \text{Var}\big(P_{\mathcal{L}}^{(a)}\big)$
      *// Try to find valid split via error pivoting*
      **while** candidates remain in $P_{\mathcal{L}}$ **do**
        Select pivot (highest-error point): $j^* = \arg\max_j \epsilon_{\mathcal{L}}[j], \quad p_s \leftarrow P_{\mathcal{L}}[j^*]$
        Partition $\mathcal{L}$ into: $\mathcal{L}_{\text{left}} = \{p \in P_{\mathcal{L}} \mid p^{(a^*)} < p_s^{(a^*)}\}, \quad \mathcal{L}_{\text{right}} = P_{\mathcal{L}} \setminus \mathcal{L}_{\text{left}}$
        **if** $\mathcal{L}_{\text{left}} \neq \emptyset$ **and** $\mathcal{L}_{\text{right}} \neq \emptyset$ **then**
          $\nu \leftarrow$ Node($\mathcal{L}_{\text{left}}, \mathcal{L}_{\text{right}}$, pivot $= j^*$, axis $= a^*$) {Pivot is a global mesh vertex index}
          Replace leaf $\mathcal{L}$ with internal node $\nu$ in $\mathcal{T}$;   **break**
        **else**
          Remove $j^*$ from candidates {Avoid empty partition}
        **end if**
      **end while**
    **end if**
  **end for**
**end for**

*// Phase 2: Bottom-up Edge Generation*

Initialize edge levels $\mathcal{E}_\ell \leftarrow \emptyset$ for all levels $\ell$
Set current level $\ell \leftarrow 1$
Initialize current node set $\mathcal{N}_{\text{curr}} \leftarrow \emptyset$
**for each** leaf $\mathcal{L}$ in $\mathcal{T}$ **do**
  **for each** index $i$ where $p_i \in \mathcal{L}$ and $\epsilon_i \geq \tau_{\text{err}}$ **do**
    Let $s \leftarrow$ parent($\mathcal{L}$).pivot
    $\mathcal{E}_1 \leftarrow \mathcal{E}_1 \cup \{(s, i)\}$ {Level 1: connect high-error nodes to local pivot}
  **end for**
  Add parent($\mathcal{L}$) to $\mathcal{N}_{\text{curr}}$
**end for**
**while** $|\mathcal{N}_{\text{curr}}| > 1$ **do**
  $\ell \leftarrow \ell + 1; \quad \mathcal{N}_{\text{next}} \leftarrow \emptyset$
  **for each** node $n \in \mathcal{N}_{\text{curr}}$ **do**
    $\mathsf{a}_k \leftarrow$ AncestorClamp($n, k$) {Clamp to root if depth($n$) $< k$}
    **if** $\mathsf{a}_k \neq n$ **then**
      $\mathcal{E}_\ell \leftarrow \mathcal{E}_\ell \cup \{(\mathsf{a}_k.\text{pivot}, n.\text{pivot})\}$ {Long-range skip connection}
      $\mathcal{N}_{\text{next}} \leftarrow \mathcal{N}_{\text{next}} \cup \{\mathsf{a}_k\}$
    **end if**
  **end for**
  $\mathcal{N}_{\text{curr}} \leftarrow$ Unique($\mathcal{N}_{\text{next}}$)
**end while**
**Return** $\mathcal{E} = \bigcup_\ell \mathcal{E}_\ell$

---

In many PDE systems, critical solution features such as shocks, vortices, or downstream wakes are spatially concentrated yet have nonlocal impact on the flow field. These regions typically incur higher prediction errors and play a disproportionate role in governing system dynamics. Motivated by this, we construct a tree that partitions the domain based on local prediction error $\epsilon$ and spatial variability of the input points $\mathbf{P}$. The goal is to generate a hierarchy of graph edges that align with the

physical importance of different regions, thus guiding the GNN to prioritize information flow where it matters most.

Algorithm 1 presents an adaptive, error-driven tree construction procedure designed to induce a physically meaningful message passing structure for Graph Neural Networks in the context of partial differential equation (PDE) prediction tasks.

Start from the root node containing all mesh points $\mathbf{P}$ and corresponding error values $\epsilon$, the algorithm recursively partitions leaves up to a depth $n_{\text{levels}}$. At each level, for each leaf, the split axis is chosen as the dimension along which the set of point coordinates in the current leaf has the highest variance, and the split point is selected as the location with the highest local error. This ensures that refinement is driven by both geometric variability and local prediction failure, making the tree sensitive to problem specific physical features.

To avoid degenerate splits, fallback strategies are used if one side of the partition is empty, by retrying the split with the next highest error point. Each successful partition produces an internal node in the binary tree, with left and right children corresponding to the resulting subsets.

After the tree is constructed, a level-wise edge extraction phase is performed. Points with error exceeding a threshold $\tau_{\text{err}}$ are first connected to their immediate parent's split index, forming the initial set of level 1 edges. Higher level edges are then generated by connecting each node to its $k$-hop ancestor in the tree hierarchy, where the $k$-*hop ancestor* of a node is defined as the node reached by traversing $k$ steps upward in the tree from the current node. This process yields a multiscale edge structure $\mathcal{E} = \bigcup_\ell \mathcal{E}_\ell$. This hierarchical edge set serves as a meaningful inductive bias for Graph Neural Networks, enabling the model to prioritize information flow in regions of high physical complexity.

Overall, this method bridges physical understanding (via error localization) and learning architecture design (via edge construction), allowing for more sample efficient and physically consistent learning in GNN-based PDE solvers.

**Undirected implementation:** Although we write an edge as an ordered pair $(i, j)$ for notational convenience, the mesh graph is treated as undirected. In implementation, we store each undirected edge by adding both directions $(i, j)$ and $(j, i)$, we apply the same convention to all augmented edges in $\mathcal{E}^\star$.

**Indexing convention:** Each leaf $\mathcal{L}$ stores a set of points with *global mesh indices* $I_\mathcal{L} \subset \{1, \ldots, N\}$. For any internal node $\nu$, $\text{pivot}(\nu) \in \{1, \ldots, N\}$ denotes the *global index* of the pivot mesh vertex. All edges returned by Algorithm 1 are edges between mesh nodes in $V$.

**Refresh convention:** The adaptive tree is rebuilt from scratch at every augmentation update. In particular, the augmentation edge set is reset as $\mathcal{E}^\star \leftarrow \emptyset$ before running Algorithm 1 on the current predicted error field. Therefore, augmentation edges are not accumulated across cycles. Previously added edges are automatically pruned whenever their associated nodes are no longer selected by the current error map. This behavior lets the model reduce extra connectivity as training improves while preserving the original mesh graph $\mathcal{E}_b$.

**Edge semantics:** The edges in $\mathcal{E}^\star$ are not intended to be a physics exact discretization of characteristic, streamline, or stress-transport directions. They are additional learned communication pathways for the neural message-passing model. Directional and anisotropic information remains available through edge attributes such as relative displacement, distance kernels, and optional geometric encodings. Thus, even though the augmented topology is undirected, the message functions can still learn direction dependent responses from the edge features.

### A.2. Architecture

MiSe-GNN follows an encoder–processor–decoder design, implemented using modular components from the `physicnemo` library (PhysicsNeMo Contributors, 2025). For each field variable (e.g., each target physical field), we instantiate a separate MiSe-GNN model.

## B. Experimental Setup

### B.1. Datasets

We evaluate on four subsets of the PLAID collection and follow its official train/test splits (Casenave et al., 2025b). The datasets consist of numerical simulations with varying mesh resolutions and topologies, inputs may include the mesh

(geometry and connectivity) and a few scalar features, while outputs comprise scalar quantities and fields defined on the native mesh. A brief summary of each subset follows (see PLAID paper for details and examples).

*Table 8.* Statistics and splits *reproduced from* PLAID (Casenave et al., 2025b). * means node count/connectivity are fixed across samples (node coordinates vary).

| Dataset | Mesh (mean nodes) | Inputs | Outputs | Splits (train/test) |
|---|---|---|---|---|
| 2D_MultiScHypEl | TRI (5,692) | MESH, 3 SCALARS | 1 SCALAR, 7 FIELDS | 764 / 376 |
| 2D_profile | TRI (37,042) | MESH | 4 FIELDS | 300 / 100 |
| VKI-LS59 | QUAD (36,421*) | MESH, 2 SCALARS | 6 SCALARS, 7 FIELDS | 671 / 168 |
| Tensile2d | TRI (9,428) | MESH, 6 SCALARS | 4 SCALARS, 6 FIELDS | 500 / 200 |

- 2D_MultiScHypEl: 2D hyperelastic homogenization of porous RVE (Representative Volume Element) under KUBC (Kinematically Uniform Boundary Conditions) (Yvonnet, 2019), solved with DOLFINx (Baratta et al., 2023). Inputs are triangular meshes and three macroscopic strain components, outputs include one global energy and seven mesh fields (displacements and first Piola–Kirchhoff stresses, plus energy density) (Casenave et al., 2025b).

- 2D_profile: 2D compressible steady RANS (Reynolds-Averaged Navier-Stokes) around deformed airfoil-like profiles, solved with elsA (Cambier, Laurent et al., 2013) using the finite volumes method. We use the cropped near profile zone, inputs are anisotropic unstructured meshes, outputs are four flow fields (Mach, Pressure, Velocity-x, Velocity-y) (Casenave et al., 2025b).

- VKI-LS59: 2D Compressible steady RANS (Reynolds-Averaged Navier-Stokes) for the VKI-LS59 blade, solved with BROADCAST (Poulain et al., 2023) by finite volumes method with high-order corrections, using a Spalart–Allmaras turbulence model. Inputs are block-structured meshes with an extra distance-to-surface field and two scalars (angle_in, mach_out), outputs are six global scalars and seven fields, with M_iso defined on the blade surface (Casenave et al., 2025b).

- Tensile2d: 2D structural mechanics in small deformation and plane strain, solved with the finite element solver Z-set (Mines ParisTech and ONERA the French aerospace lab, 1981-present). A slab is loaded by negative pressure on the top and clamped at the bottom, inputs are the unstructured mesh plus one load and five material parameters, outputs are four global scalars and six stress/displacement fields (Casenave et al., 2025b).

## B.2. Baselines

This appendix summarizes the baselines considered in our study. Descriptions and configuration choices are aligned with, and should be cited to, the PLAID benchmark (Casenave et al., 2025b), complemented by the original references.

**MeshGraphNets (MGN) (Pfaff et al., 2021a):** an encode–process–decode GNN that converts meshes into graphs, propagates information via message passing, and decodes per-node fields and scalar observables.

**MGN Tree (Ripken et al., 2023):** a multiscale extension that introduces long-range, tree-derived edges (from octree embeddings) to shorten graph diameter and accelerate global information flow.

**Mesh Morphing Gaussian Processes (MMGP) (Casenave et al., 2024):** a kernel pipeline that pre-processes mesh-based data via morphing, finite-element interpolation, and dimensionality reduction into a compact latent, followed by Gaussian processes to predict scalars and fields.

**Vi-Transformer (Dosovitskiy et al., 2021)** and **Augur**[1]**:** both form tokens from mesh partitions (capturing local neighborhoods) and use a transformer backbone to produce scalar and field predictions.

---

[1]Commercial solution by Augur.

**Fourier Neural Operator (FNO) (Li et al., 2021):** an operator-learning baseline that maps input fields to solution fields via stacked *Fourier layers*. Each layer performs a global spectral convolution by applying an Fast Fourier Transform (FFT), multiplying a truncated set of Fourier modes with learned complex weights, and transforming back with an inverse FFT, followed by pointwise channel mixing and nonlinearities, the network lifts to a latent width and finally projects to the output field.

**MARIO (Catalani et al., 2025) (cf. (Catalani et al., 2024)):** an implicit representation approach (building on (Catalani et al., 2024)) that models continuous fields via coordinate-based networks and incorporates resolution-invariance for aerodynamics.

### B.3. Training Protocol

**Optimization:** All models are trained with the Adam optimizer using learning rate $10^{-3}$, no learning rate scheduler, batch size 1, and 1000 epochs. We use a mean squared error objective and LeakyReLU activations throughout the network. Model selection is performed by tracking validation loss over epochs and saving the checkpoint that achieves the lowest validation loss for each model.

For MiSe-GNN, the error target $\epsilon^\star$ is detached as described in Eq. (5). Consequently, gradients from the error loss update the predicted error head and the shared encoder-processor, but do not alter $\hat{\mathbf{y}}_{\mathrm{base}}$ or the field head through the target construction. Unless otherwise stated, no additional gradient stopping is applied before the error head. Formally, the training loss is:

$$\mathcal{L}_{\mathrm{MSE}} = \frac{1}{N}\sum_{i=1}^{N}\big\|\hat{\mathbf{y}}_i - \mathbf{y}_i\big\|_2^2 \tag{9}$$

where $\hat{\mathbf{y}}_i$ and $\mathbf{y}_i$ denote the predicted and reference targets, respectively.

**Hardware and datasets:** Unless stated otherwise, each experiment is run on a single GPU. The dataset-hardware mapping is:

*Table 9.* Training hardware used per dataset.

| Dataset | GPU |
| --- | --- |
| `2D_MultiScHypEl`, `Tensile2d` | 1 NVIDIA A30 |
| `2D_profile`, `VKI-LS59` | 1 NVIDIA A100 |

**Feature inputs:** The base node/edge inputs used across all experiments are defined once in Appendix C.1 and are augmented by the geometric features introduced there.

**Reproducibility notes:** Unless otherwise specified elsewhere in the paper, unspecified hyperparameters follow the defaults of the underlying `physicnemo` modules.

## C. Geometric Feature Engineering

**Notation:** Let $G = (V, \mathcal{E})$ be the graph built on a 2D mesh of a domain $\Omega \subset \mathbb{R}^2$ with boundary $\partial\Omega$. Each node $i \in V$ has coordinates $\mathbf{x}_i \in \mathbb{R}^2$ and base attributes (node type $t_i$, boundary distance $d_i$). For an edge $(i, j) \in \mathcal{E}$, define

$$\Delta\mathbf{x}_{ij} := \mathbf{x}_j - \mathbf{x}_i, \qquad r_{ij} := \|\Delta\mathbf{x}_{ij}\|_2, \qquad \mathbf{u}(\theta) := [\cos\theta,\ \sin\theta]^\top$$

### C.1. Base Features

**Node features (base):** We use spatial coordinates, node-type indicator, and distance to boundary:

$$\mathbf{z}_i^{\mathrm{base}} = \big[\,\mathbf{x}_i,\ t_i,\ d_i\,\big] \text{ (concatenated with any available input scalars)}$$

**Edge features (base):** Edges include the relative displacement together with a distance-based kernel weight computed via a bandwidth $h$:

$$w_{ij} \; = \; \exp\!\left(-\frac{r_{ij}^2}{2h^2}\right), \qquad h \; = \; \mathrm{median}\big\{\, \|\mathbf{x}_p - \mathbf{x}_q\|_2 : (p,q) \in \mathcal{E}_b \,\big\} \tag{10}$$

and we form

$$\mathbf{e}_{ij}^{\mathrm{base}} \; = \; \big[\; w_{ij}, \; \Delta\mathbf{x}_{ij} \;\big]$$

These base inputs are used when no additional feature engineering is applied, the constructions below are optional augmentations.

**Consistency for Augmented Edges:** For any augmented edge $(i,j) \in \mathcal{E}^\star$, we compute its edge attributes using the *same* feature construction as for base mesh edges. We compute $h$ once from the base mesh edges $\mathcal{E}_b$ and keep it fixed when updating $\mathcal{E}^\star$. This consistency extends to all optional geometric augmentations (e.g., Sinus, Sph, see Appendix C.2) or domain-specific flags (see Appendix C.3), which are derived solely from endpoint coordinates. This ensures that the GNN processes local physical edges and long-range skip connections within a unified feature space.

## C.2. Geometric Encodings

### C.2.1. DIRECTIONAL INTEGRATED DISTANCE (DID)

**Boundary distance:** The distance from $\mathbf{x} \in \Omega$ to the boundary along direction $\theta$ is

$$g(\mathbf{x}, \theta) \; := \; \inf\{\, r \geq 0 : \; \mathbf{x} + r\,\mathbf{u}(\theta) \in \partial\Omega \,\}, \qquad \text{with } g(\mathbf{x}, \theta) := R_{\max} \text{ if the set is empty.} \tag{11}$$

**DID over an angular sector:** Given a sector $[\theta_j, \theta_j']$ and a nonnegative weight $w_j(\theta)$ (e.g., uniform or Gaussian on the sector), the *Directional Integrated Distance* feature at $\mathbf{x}$ for that sector is

$$\mathrm{DID}_j(\mathbf{x}) \; := \; \int_{\theta_j}^{\theta_j'} w_j(\theta)\, g(\mathbf{x}, \theta)\, \mathrm{d}\theta \tag{12}$$

Using $J$ (possibly overlapping) sectors yields the vector $\big(\mathrm{DID}_1(\mathbf{x}), \ldots, \mathrm{DID}_J(\mathbf{x})\big)$.

**Discrete approximation:** With $K_j$ sample angles $\{\theta_{jk}\}_{k=1}^{K_j} \subset [\theta_j, \theta_j']$ and normalized weights $\tilde{w}_{jk} := \frac{w_j(\theta_{jk})}{\sum_{k'=1}^{K_j} w_j(\theta_{jk'})}$,

$$\widehat{\mathrm{DID}}_j(\mathbf{x}) \; := \; \sum_{k=1}^{K_j} \tilde{w}_{jk}\, g(\mathbf{x}, \theta_{jk}), \qquad g(\mathbf{x}, \theta_{jk}) \text{ as in (11)} \tag{13}$$

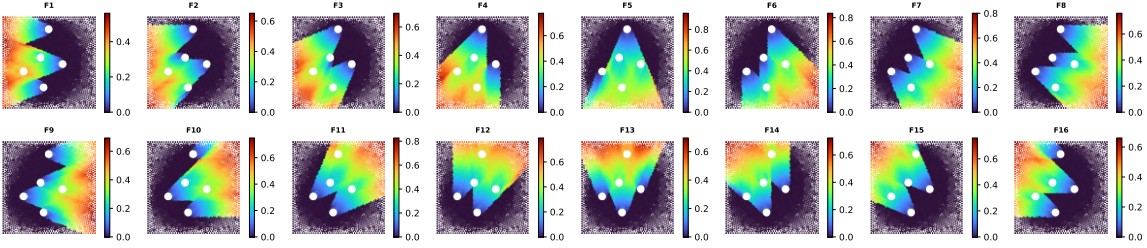

*Figure 8.* Directional Integrated Distance (DID) (`2D_MultiScHypEl` dataset).

**Use in a GNN:** Augment node $i$ with

$$\mathbf{z}_i^{\mathrm{geom}} = \Big[\mathrm{DID}_1(\mathbf{x}_i), \ldots, \mathrm{DID}_J(\mathbf{x}_i)\Big]$$

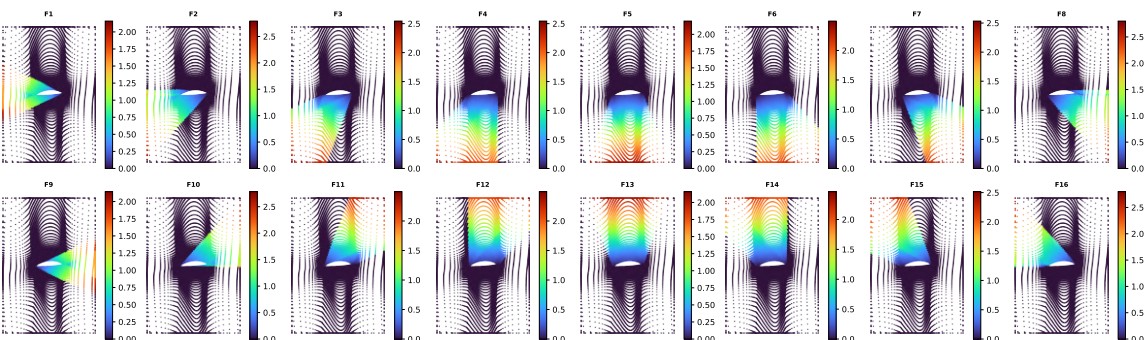

*Figure 9.* Directional Integrated Distance (DID) (`2D_profile` dataset).

### C.2.2. SINUSOIDAL BASIS (SINUS)

For any $u \in \mathbb{R}$ (e.g., a Cartesian coordinate or a node boundary distance), define a $2M$ dimensional multi-frequency sinusoidal embedding:

$$\text{PE}_M(u) := \big[\, \sin(\omega_1 u),\, \cos(\omega_1 u),\, \ldots,\, \sin(\omega_M u),\, \cos(\omega_M u) \,\big] \tag{14}$$

A convenient choice is a geometric ladder of frequencies spanning a spatial band:

$$\omega_i = \omega_{\min} \left( \frac{\omega_{\max}}{\omega_{\min}} \right)^{\frac{i-1}{M-1}}, \quad i = 1, \ldots, M, \qquad \omega_{\min} \approx \frac{1}{L}, \;\; \omega_{\max} \approx \frac{c}{s} \tag{15}$$

where $L$ is a domain length scale, $s$ a characteristic mesh spacing, and $c$ a dimensionless factor (e.g., $c = \frac{4L}{\pi s}$ as suggested in practice) [2].

For a 2D point $\mathbf{x} = (\mathbf{x}^{(1)}, \mathbf{x}^{(2)})$, use $\text{PE}_M(\mathbf{x}) := \text{PE}_M(\mathbf{x}^{(1)}) \,\|\, \text{PE}_M(\mathbf{x}^{(2)})$, and analogously for distances $d$.

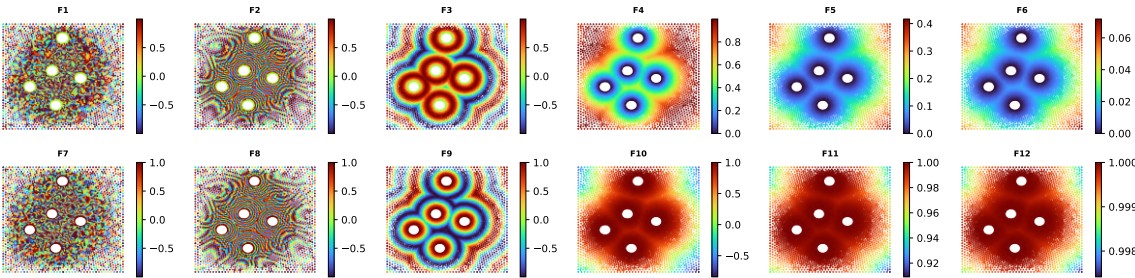

*Figure 10.* Sinusoidal Basis Distance to Holes (`2D_MultiScHypEl` dataset).

**Use in a GNN:** Augment node features with sinusoidal embeddings of selected scalars (e.g., $\mathbf{x}^{(1)}$, $\mathbf{x}^{(2)}$, distance to boundary $d$), and augment edge $(i, j)$ with embeddings of

$$[\text{PE}_M(\Delta \mathbf{x}_{ij}^{(1)}), \text{PE}_M(\Delta \mathbf{x}_{ij}^{(2)}), \text{PE}_M(r_{ij})]$$

### C.2.3. SPHERICAL HARMONICS BASIS (SPH) FOR 2D ANGLES

Let $\theta \in [-\pi, \pi]$ be an angle (e.g., the polar angle of $\mathbf{x}$ about a reference point, or the angle of an edge vector). Using the real $m = 0$ spherical harmonics restricted to 2D angles gives the basis:

$$Y_\ell^0(\theta) := c_\ell\, P_\ell(\cos\theta), \qquad c_\ell = \sqrt{\frac{(2\ell + 1)!}{4\pi}}, \qquad \ell = 1, \ldots, L \tag{16}$$

---

[2] Any reasonable band that covers coarse to fine scale spatial variation works, (15) is a pragmatic setting.

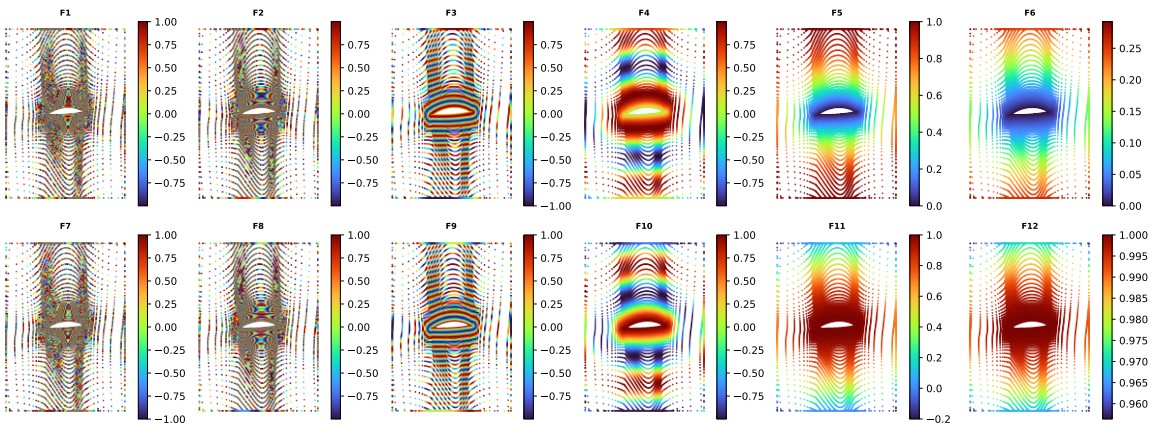

*Figure 11.* Sinusoidal Basis Distance to Airfoil (2D_profile dataset).

where $P_\ell$ is the Legendre polynomial of degree $\ell$. Because $Y_\ell^0$ is even in $\theta$, it cannot distinguish $\theta$ from $-\theta$. To disambiguate orientation, introduce the companion (odd) basis:

$$\overline{Y}_\ell^0(\theta) := c_\ell P_\ell(\sin\theta), \qquad \ell = 1, \ldots, L \tag{17}$$

We then define the angular embedding:

$$\mathrm{SPH}_L(\theta) := \left[ Y_1^0(\theta), \ldots, Y_L^0(\theta), \overline{Y}_1^0(\theta), \ldots, \overline{Y}_L^0(\theta) \right] \in \mathbb{R}^{2L} \tag{18}$$

**Use in a GNN:** For each angle of interest, the angle of $\mathbf{x}$ in a chosen coordinate frame, or the edge angle $\theta_{ij} = \mathrm{atan2}\left((\Delta\mathbf{x}_{ij})^{(2)}, (\Delta\mathbf{x}_{ij})^{(1)}\right)$, append $\mathrm{SPH}_L(\cdot)$ to the corresponding node/edge feature vector. Multiple frames (e.g., centered at distinct geometry landmarks) can be handled by concatenating (18) computed in each frame.

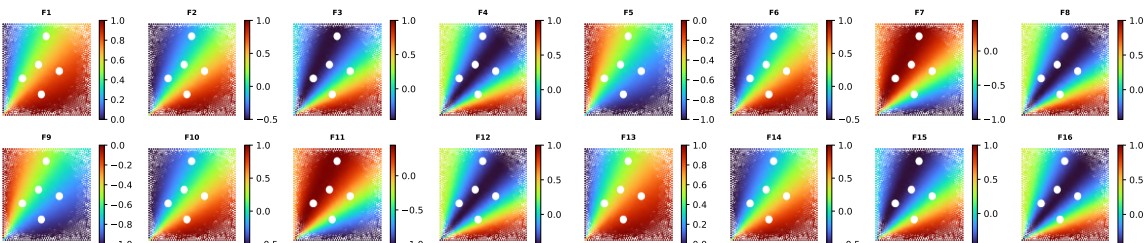

*Figure 12.* Spherical Harmonics of Angles in the Original Coordinate Frame (2D_MultiScHypEl dataset).

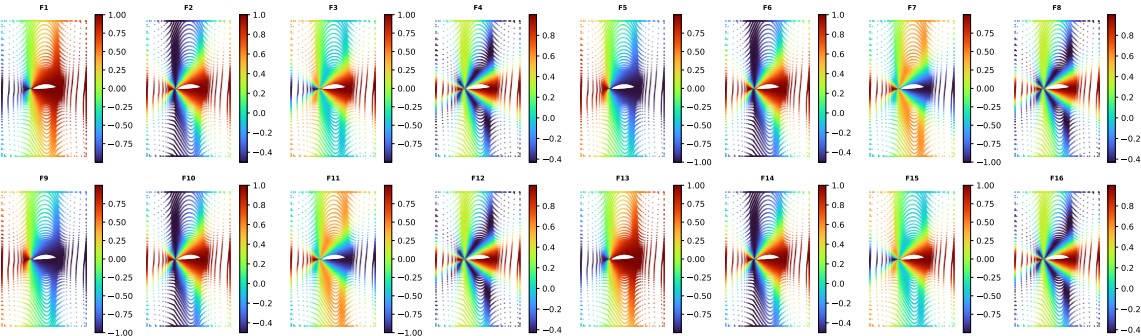

*Figure 13.* Spherical Harmonics of Angles in the Original Coordinate Frame (2D_profile dataset).

C.2.4. COMPACT RECIPE FOR THE MODEL INPUT

A practical, expressive node input is:

$$\mathbf{z}_i^{(0)} \;=\; \left[\; \mathbf{z}_i^{\text{base}} \;;\; \mathrm{DID}_1(\mathbf{x}_i), \ldots, \mathrm{DID}_J(\mathbf{x}_i) \;;\; \mathrm{PE}_M(d_i) \;;\; \mathrm{SPH}_L(\theta_i) \;\right] \tag{19}$$

with $\theta_i$ any relevant angle for node $i$ (e.g., in a chosen polar frame). For edges, use:

$$\mathbf{e}_{ij} \;=\; \left[\; \mathbf{e}_{ij}^{\text{base}} \;;\; \mathrm{PE}_M(r_{ij}) \;;\; \mathrm{SPH}_L(\theta_{ij}) \;\right] \tag{20}$$

**Hyperparameters used in experiments:** Table 10 reports the hyperparameters used to instantiate DID, Sinus, and Sph features.

*Table 10.* Hyperparameters for generic geometric encodings used in all experiments.

| Encoding | Hyperparameter | Meaning | Value |
|---|---|---|---|
| DID | $J$ | # angular sectors | 16 |
| | $\{\theta_j, \theta_j'\}$ | Sector definition (uniform partition of $[-\pi, \pi)$) | $\theta_j = -\pi + \frac{2\pi(j-1)}{J}$, $\theta_j' = \theta_j + \frac{2\pi}{J}$ |
| | $w_j(\theta)$ | Sector weight | uniform |
| | $K$ | # features per sector | 1 (Mean distance) |
| | $\theta_{\text{rot}}$ | Sector center step $\Rightarrow J = \lceil 2\pi/\theta_{\text{rot}} \rceil = 16$ | $\pi/8$ |
| | $\theta_{\text{seg}}$ | Sector width (overlap since $\theta_{\text{seg}} > \theta_{\text{rot}}$) | $\pi/4$ |
| | $R_{\max}$ | Distance clamp | 4 |
| Sinus | $M$ | # frequencies (output dim $2M$) | 6 |
| | $\omega_{\min}$ | Minimum frequency ($1/L_{\text{dom}}$) | 1/6 |
| | $\omega_{\max}$ | Maximum frequency ($c/s$) | 1000 |
| | $c$ | Scale factor in $\omega_{\max}$ | 1 |
| | $L_{\text{dom}}, s$ | Domain scale and mesh spacing estimator | 6, $10^{-3}$ |
| Sph | $L_{\text{sph}}$ | Max degree (Legendre polynomial order) | 4 |
| | frames | Angle reference frame(s) used | 4 planar projections |

## C.3. Domain Specific Features

**Hyperparameters for OurFeat:** Table 11 summarizes the exact thresholds and settings used to extract domain-specific features.

*Table 11.* Hyperparameters for domain-specific features (**OurFeat**).

| Dataset | Hyperparameter | Meaning | Value |
|---|---|---|---|
| `2D_MultiScHypEl` | $K_{\text{hole}}$ | # nearest holes in $\mathrm{Polar}(\mathbf{x}_i) \in \mathbb{R}^{3K_{\text{hole}}}$ | 4 |
| | $R$ | Porosity probe radius $R = \alpha\bar{r}$ | $\alpha = 5$ |
| | $\varepsilon$ | Stability constant in $\rho_i = \frac{d_{1,i}}{d_{2,i}+\varepsilon}$ | $10^{-6}$ |
| `Tensile2d` | – | Top/bottom filtering margin when extracting $\Gamma_L, \Gamma_R$ | 0 |
| | – | Corner filtering margin | $10^{-8}$ |

C.3.1. 2D_MULTISCHYPEL

For the `2D_MultiScHypEl` dataset the domain contains several circular holes. We exploit this structure by first recovering approximate hole centers and radii from the mesh, and then building node features that encode the relative position of each node with respect to these holes.

**Recovering hole centers:** From the nodal tag `Holes` we collect all boundary nodes $\{\mathbf{x}_j\}_{j \in \mathcal{H}} \subset \mathbb{R}^2$ and cluster them into connected components, each corresponding to one hole. For each component $h$ we fit a circle to its boundary nodes by least squares, obtaining a center $\mathbf{c}_h \in \mathbb{R}^2$ and radius $r_h > 0$. We denote the set of holes by $\mathcal{H}_{\text{holes}}$ and the distance from node $i$ to hole center $h$ by

$$D_{ih} := \left\| \mathbf{x}_i - \mathbf{c}_h \right\|_2, \qquad h \in \mathcal{H}_{\text{holes}}$$

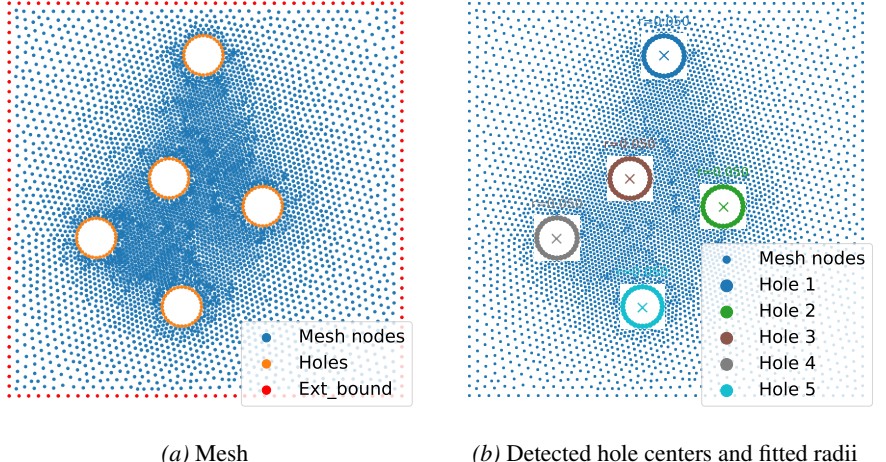

*(a)* Mesh          *(b)* Detected hole centers and fitted radii

*Figure 14.* Recovering approximate hole centers and radii from the mesh (2D_MultiScHypEl dataset).

**K–nearest polar coordinates (Polar):** For each node $\mathbf{x}_i$ we select the $K$ nearest hole centers $\mathcal{N}_K(i) \subset \mathcal{H}_{\text{holes}}$ according to $D_{ih}$. For each $h \in \mathcal{N}_K(i)$ we define the displacement $\boldsymbol{\delta}_{ih} := \mathbf{x}_i - \mathbf{c}_h$, the radial distance $r_{ih} := \|\boldsymbol{\delta}_{ih}\|_2$ and the corresponding polar angle $\theta_{ih} := \operatorname{atan2}\big(\boldsymbol{\delta}_{ih}^{(2)}, \boldsymbol{\delta}_{ih}^{(1)}\big)$. The polar embedding of node $i$ is

$$\operatorname{Polar}(\mathbf{x}_i) := \big[r_{ih},\, \cos\theta_{ih},\, \sin\theta_{ih}\big]_{h \in \mathcal{N}_K(i)} \in \mathbb{R}^{3K}$$

which encodes the relative position of $\mathbf{x}_i$ with respect to its nearest holes (Figure 15).

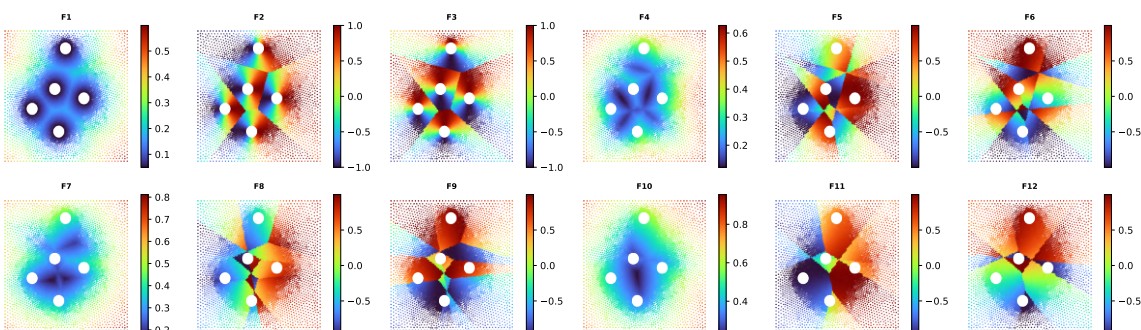

*Figure 15.* Polar embedding $\operatorname{Polar}(\mathbf{x}_i)$: radial distance and angle of each node with respect to its nearest hole centers (2D_MultiScHypEl dataset).

**Local porosity and nearest hole gap:** Let $\bar{r}$ be the mean radius of all holes, and set a probe scale $R := 5\,\bar{r}$. We approximate the local porosity around node $i$ as the fraction of the disk $B(\mathbf{x}_i, R)$ covered by the holes:

$$p_i := \frac{1}{R^2} \sum_{h \in \mathcal{H}_{\text{holes}}} \mathbf{1}[D_{ih} \leq R]\, r_h^2$$

where $\mathbf{1}[\cdot]$ is the indicator function (Figure 16a).

For each node we sort the distances $\{D_{ih}\}_{h \in \mathcal{H}_{\text{holes}}}$ and denote the first two order statistics by $d_i^{(1)} \leq d_i^{(2)}$. We then define

$$d_{1,i} := d_i^{(1)}, \qquad d_{2,i} := d_i^{(2)}, \qquad \rho_i := \frac{d_{1,i}}{d_{2,i} + \varepsilon}$$

with a small $\varepsilon > 0$ for numerical stability (Figure 16). Here $d_{1,i}$ and $d_{2,i}$ measure proximity to the nearest and second–nearest holes, while $\rho_i$ highlights regions where two holes are at comparable distance (narrow gaps).

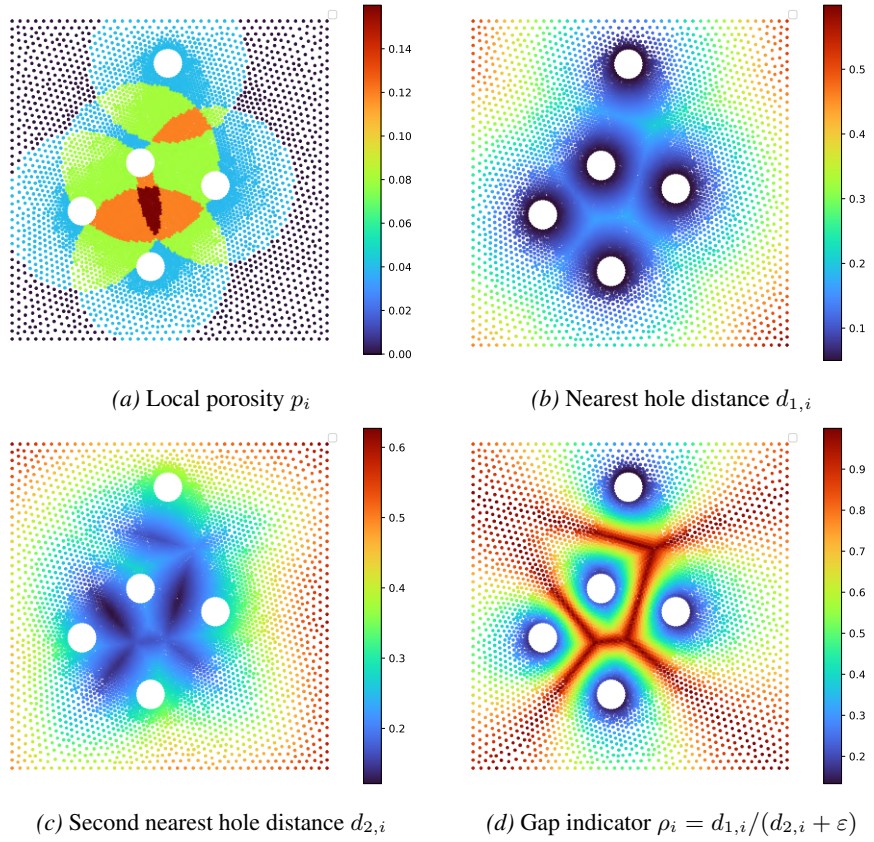

*(a)* Local porosity $p_i$

*(b)* Nearest hole distance $d_{1,i}$

*(c)* Second nearest hole distance $d_{2,i}$

*(d)* Gap indicator $\rho_i = d_{1,i}/(d_{2,i} + \varepsilon)$

*Figure 16.* Local porosity and nearest hole gap.

**Sinusoidal embedding of the second nearest distance** ($\mathrm{Sinus}(d_{2,i})$)**:** To capture oscillatory dependence on the inter hole spacing, we apply the sinusoidal basis from Appendix C.2.2 to the second nearest distance $d_{2,i}$ (Figure 17).

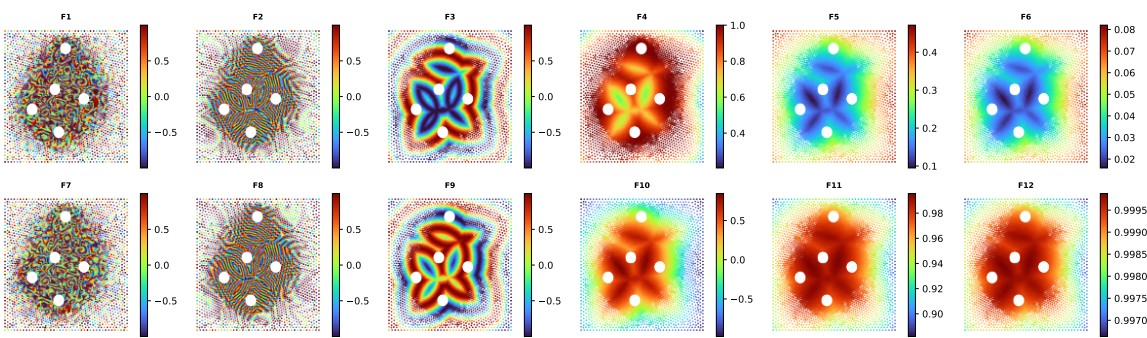

*Figure 17.* Sinus(F4) (`2D_MultiScHypEl` dataset).

**Resulting node and edge input:** For this dataset we augment the base node input from Appendix C.1 with the hole-aware features:

$$\mathbf{z}_i^{(0)} = \left[ \mathbf{z}_i^{\mathrm{base}}; \ \mathrm{Polar}(\mathbf{x}_i); \ p_i; \ d_{1,i}; \ d_{2,i}; \ \rho_i; \ \mathrm{Sinus}(d_{2,i}) \right]$$

which provides the GNN with explicit information about local porosity, inter hole gaps and multi-scale distances to the surrounding holes.

We also enrich the edge features with two hole aware terms. For an edge $(i, j)$ with endpoints $\mathbf{x}_i, \mathbf{x}_j \in \mathbb{R}^2$ we consider the

segment

$$S_{ij}(t) := \mathbf{x}_i + t\,(\mathbf{x}_j - \mathbf{x}_i), \qquad t \in [0,1]$$

and, for each hole $h$ with center $\mathbf{c}_h$ and radius $r_h$, we project $\mathbf{c}_h$ onto this segment. Let

$$t_h^* := \mathrm{clip}_{[0,1]}\left(\frac{(\mathbf{c}_h - \mathbf{x}_i)^\top (\mathbf{x}_j - \mathbf{x}_i)}{\|\mathbf{x}_j - \mathbf{x}_i\|_2^2}\right), \qquad d_h^{(ij)} := \left\|S_{ij}(t_h^*) - \mathbf{c}_h\right\|_2$$

The *occlusion* indicator marks whether the edge segment intersects any hole disk:

$$\mathbf{e}_{ij}^{\mathrm{occ}} := \mathbf{1}\left[\exists h \in \mathcal{H}_{\mathrm{holes}} \text{ such that } d_h^{(ij)} \leq r_h\right]$$

In addition, we encode whether the two endpoints of the edge share the same nearest hole. Let

$$h_i := \arg\min_{h \in \mathcal{H}_{\mathrm{holes}}} \|\mathbf{x}_i - \mathbf{c}_h\|_2, \qquad h_j := \arg\min_{h \in \mathcal{H}_{\mathrm{holes}}} \|\mathbf{x}_j - \mathbf{c}_h\|_2$$

and define the *shared nearest hole* flag

$$\mathbf{e}_{ij}^{\mathrm{same}} := \mathbf{1}\left[h_i = h_j\right]$$

Starting from the base edge input $\mathbf{e}_{ij}^{\mathrm{base}}$ defined in Appendix C.1, we finally use

$$\mathbf{e}_{ij}^{(0)} = \left[\mathbf{e}_{ij}^{\mathrm{base}};\ \mathbf{e}_{ij}^{\mathrm{occ}};\ \mathbf{e}_{ij}^{\mathrm{same}}\right]$$

so that the GNN can distinguish edges that are blocked by a hole and edges whose endpoints lie in the same pore region.

*Table 12.* `2D_MultiScHypEl`, capacity scaling and feature engineering (15 MP / hidden size 80).

| Model | u1 | u2 | P11 | P12 | P22 | P21 | psi |
|---|---|---|---|---|---|---|---|
| MiSe-GNN | 0.01464 | 0.01503 | 0.02174 | 0.04812 | 0.02181 | 0.04725 | 0.03397 |
| MiSe-GNN (DID) | 0.01494 | 0.01673 | 0.01928 | 0.03620 | 0.01878 | 0.03524 | 0.03449 |
| MiSe-GNN (DID+Sinus+Sph) | 0.01555 | 0.01668 | 0.02167 | 0.03654 | 0.02321 | 0.03713 | 0.03394 |
| MiSe-GNN (OurFeat) | 0.01248 | 0.01445 | 0.01851 | 0.03814 | 0.01864 | 0.03883 | 0.03473 |
| **MiSe-GNN (OurFeat+DID)** | **0.01203** | 0.01450 | **0.01730** | **0.03119** | **0.01764** | **0.03140** | **0.03282** |
| MiSe-GNN (OurFeat+DID+Sinus) | 0.01274 | **0.01403** | 0.01763 | 0.03184 | 0.01781 | 0.03198 | 0.03333 |
| MiSe-GNN (OurFeat+DID+Sinus+Sph) | 0.01260 | 0.01465 | 0.01780 | 0.03188 | 0.01773 | 0.03208 | 0.03347 |

### C.3.2. TENSILE2D

For the `Tensile2d` dataset the domain is a tensile specimen with a symmetric constriction. We first recover approximate left and right boundary curves and then define node and edge features that encode the position of each node relative to these curves and to the throat section.

**Left/right boundary curves:**  From the triangular surface mesh we extract border edges (edges incident to exactly one triangle) and collect all boundary nodes $\{\mathbf{x}_i\}_{i \in \mathcal{B}}$. We then split these into a left set (nodes with $\mathbf{x}_i^{(1)} < 0$) and a right set (nodes with $\mathbf{x}_i^{(1)} > 0$), and discard nodes close to the top/bottom and near the extreme corners. The remaining points are ordered and treated as two polylines:

$$\Gamma_L, \Gamma_R \subset \mathbb{R}^2$$

representing the left and right curved boundaries of the specimen (Figure 18).

**Node features:**  For each mesh node $\mathbf{x}_i$ we compute its closest points on the left and right curves, $\boldsymbol{\ell}_i \in \Gamma_L$ and $\mathbf{r}_i \in \Gamma_R$, by projecting onto all line segments of each polyline (Figure 19). Let

$$d_{L,i} := \|\mathbf{x}_i - \boldsymbol{\ell}_i\|_2, \qquad d_{R,i} := \|\mathbf{x}_i - \mathbf{r}_i\|_2$$

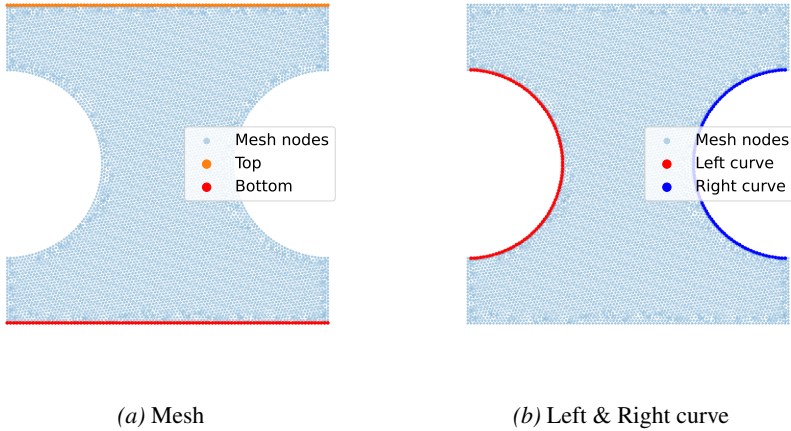

*(a)* Mesh          *(b)* Left & Right curve

*Figure 18.* Left and right boundary curves extracted from the `Tensile2d` mesh.

and define the local specimen width and a normalized lateral coordinate by

$$W_i := d_{L,i} + d_{R,i}, \qquad \eta_i := \frac{d_{L,i} - d_{R,i}}{d_{L,i} + d_{R,i}}$$

Thus $\eta_i \approx -1$ near the left boundary, $\eta_i \approx +1$ near the right boundary, and $\eta_i \approx 0$ close to the midline (Figure 20).

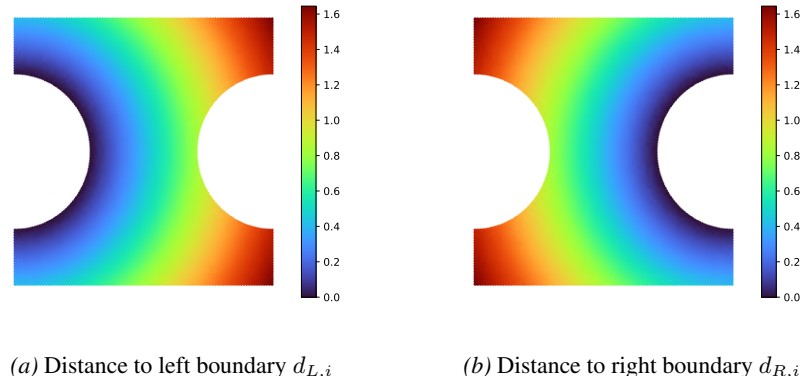

*(a)* Distance to left boundary $d_{L,i}$         *(b)* Distance to right boundary $d_{R,i}$

*Figure 19.* Distances from each node to the left and right boundary curves (`Tensile2d` dataset).

We also construct an approximate centerline by taking

$$\mathbf{c}_i := \tfrac{1}{2}(\boldsymbol{\ell}_i + \mathbf{r}_i)$$

and performing a PCA on $\{\mathbf{c}_i\}$ to obtain a mean $\boldsymbol{\mu}$ and orthonormal directions $\mathbf{e}_x, \mathbf{e}_y \in \mathbb{R}^2$ (tangent and normal to the centerline). Projecting $\mathbf{c}_i$ into this frame gives

$$s_i := (\mathbf{c}_i - \boldsymbol{\mu})^\top \mathbf{e}_x, \qquad n_i := (\mathbf{c}_i - \boldsymbol{\mu})^\top \mathbf{e}_y$$

We identify the throat as the location with minimal width $i^\star := \arg\min_i W_i$ and define an axial distance to the throat

$$\tau_i := \left| s_i - s_{i^\star} \right|$$

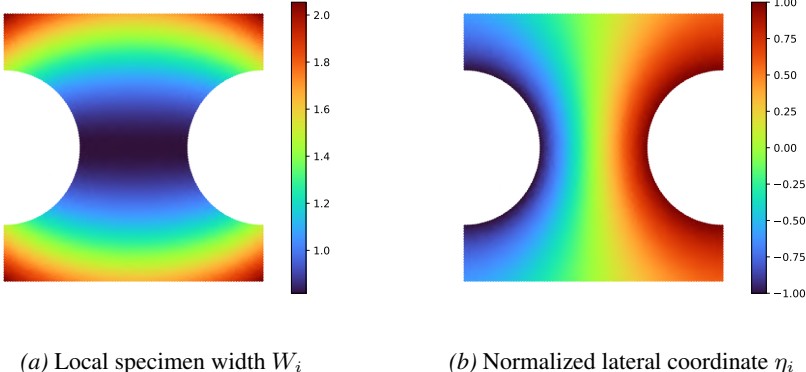

*(a)* Local specimen width $W_i$        *(b)* Normalized lateral coordinate $\eta_i$

*Figure 20.* Width and normalized lateral position of each node with respect to the left/right boundaries (`Tensile2d` dataset).

Starting from the base node input $\mathbf{z}_i^{\text{base}}$ (Appendix C.1), we augment each node with the following scalar channels:

$$\mathbf{z}_i^{(0)} = \left[\mathbf{z}_i^{\text{base}};\; d_{L,i};\; d_{R,i};\; s_i;\; n_i;\; \eta_i;\; W_i;\; \tau_i\right]$$

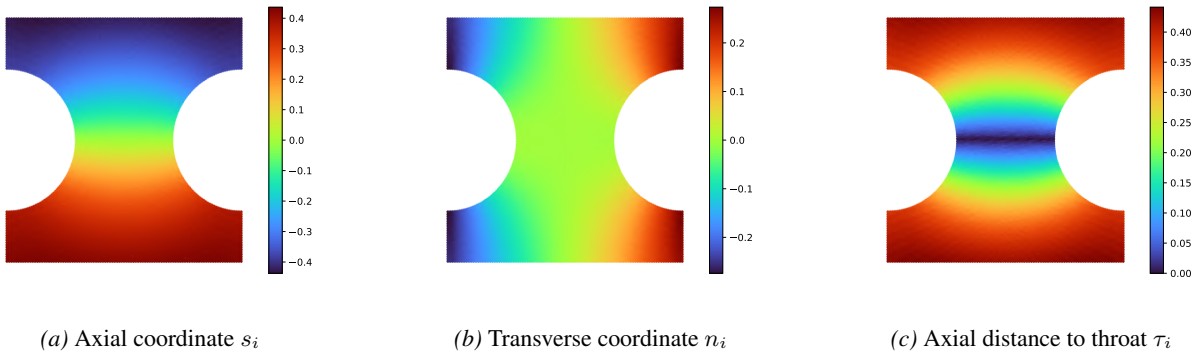

*(a)* Axial coordinate $s_i$      *(b)* Transverse coordinate $n_i$      *(c)* Axial distance to throat $\tau_i$

*Figure 21.* Centerline based coordinates for the `Tensile2d` specimen: axial position $s_i$, transverse position $n_i$, and axial distance to the throat $\tau_i$.

**Edge features:** For an edge $(i,j)$ with endpoints $\mathbf{x}_i, \mathbf{x}_j$, to encode how the edge connects regions across the width of the specimen we include differences of the nodewise coordinates:

$$d\eta_{ij} := \eta_i - \eta_j, \qquad dW_{ij} := W_i - W_j$$

We also measure the alignment of the edge with the tensile direction. Let $\mathbf{t}^{\text{ctr}} \in \mathbb{R}^2$ be the unit tangent of the centerline (taken as $\mathbf{e}_x$), and

$$\hat{\mathbf{u}}_{ij} := \frac{\Delta\mathbf{x}_{ij}}{\|\Delta\mathbf{x}_{ij}\|_2}$$

then

$$\cos_{\text{ctr},ij} := \hat{\mathbf{u}}_{ij}^\top \mathbf{t}^{\text{ctr}}$$

is the cosine of the angle between the edge and the specimen axis.

Starting from the base edge input $\mathbf{e}_{ij}^{\text{base}}$ (Appendix C.1), we use

$$\mathbf{e}_{ij}^{(0)} = \left[\mathbf{e}_{ij}^{\text{base}};\; d\eta_{ij};\; dW_{ij};\; \cos_{\text{ctr},ij}\right]$$

which allows the GNN to distinguish edges by their geometric scale, their position across the width, and their orientation relative to the tensile direction.

*Table 13.* `Tensile2d`, capacity scaling and feature engineering (15 MP / hidden size 80).

| Model | U1 | U2 | sig11 | sig22 | sig12 |
|---|---|---|---|---|---|
| MiSe-GNN | 0.00402 | 0.00609 | 0.00597 | 0.00178 | 0.00226 |
| **MiSe-GNN (Sinus+Sph)** | 0.00343 | 0.00428 | **0.00466** | **0.00133** | **0.00157** |
| MiSe-GNN (OurFeat) | 0.00328 | 0.00432 | 0.00523 | 0.00159 | 0.00219 |
| MiSe-GNN (OurFeat+Sinus+Sph) | **0.00289** | **0.00402** | 0.00497 | 0.00160 | 0.00231 |

# D. Additional Benchmark Results

### D.1. `VKI-LS59` dataset

**Base setting (10 MP / hidden size 32):** Table 14 reports the RRMSE for both `nut` and `mach` fields. **MiSe-GNN** achieves a significant reduction in error, outperforming both **MGN** and **MGN Tree** by large margins. Specifically, on the challenging `nut` field, **MiSe-GNN** reduces error by **60.6**% and **66.0**% relative to **MGN** and **MGN Tree**, respectively. For the `mach` field, **MiSe-GNN** achieves the lowest error, with improvements of **36.1**% and **7.6**% compared to the baselines. See Figure 22 for qualitative comparisons on field `mach`.

These results demonstrate that MiSe-GNN generalizes robustly across industrial-type CFD cases, confirming the method's flexibility.

*Table 14.* Results on `VKI-LS59` (10 MP / hidden size 32).

| Model | nut | mach |
|---|---|---|
| MGN | 0.1878 | 0.0344 |
| MGN Tree | 0.2177 | 0.0238 |
| **MiSe-GNN** | **0.0740** | **0.0220** |
| $\Delta_{\text{MGN}}$ | 60.60% | 36.05% |
| $\Delta_{\text{MGN Tree}}$ | 66.01% | 7.56% |

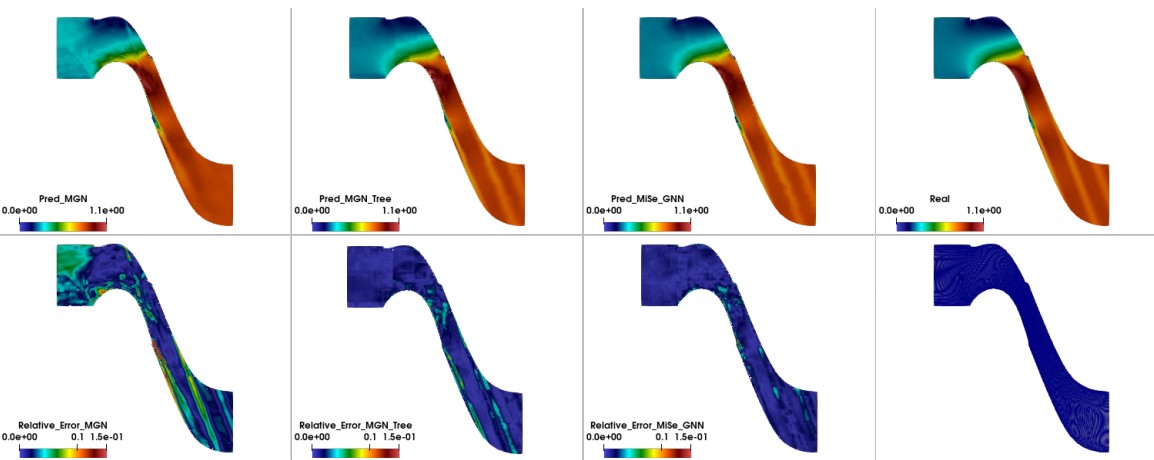

*Figure 22.* Qualitative comparison on `VKI-LS59`. Top row: predicted fields, bottom row: relative errors. From left to right: **MGN**, **MGN Tree**, **MiSe-GNN**, and ground truth.

**Capacity scaling & feature engineering (15 MP / hidden size 80):** Feature engineering primarily benefits the `mach` field. With (15/80) and *DID+Sinus+Sph*, `mach` improves from **0.01913** to **0.01558** ($-$**18.5**%), while `nut` shows little to

no improvement relative to the 15/80 base (Table 15) and remains higher than the 10/32 setting. This suggests `nut` may require different normalization or a tailored training schedule, whereas `mach` benefits from harmonic and sinusoidal bases.

*Table 15.* Results on `VKI-LS59` with capacity scaling and feature engineering (15 MP / hidden size 80).

| Model | nut | mach |
|---|---|---|
| MiSe-GNN | 0.15478 | 0.01913 |
| MiSe-GNN (DID) | 0.15340 | 0.01772 |
| **MiSe-GNN (DID+Sinus+Sph)** | **0.15207** | **0.01558** |
| MGN (DID+Sinus+Sph) | 0.15415 | 0.03099 |
| MGN Tree (DID+Sinus+Sph) | 0.15425 | 0.01908 |

To disentangle the effect of feature encodings from the proposed error-guided augmentation, we additionally equip **MGN** and **MGN Tree** with the same *DID+Sinus+Sph* features under the (15 MP / hidden size 80) setting. As shown in Table 15, **MiSe-GNN (*DID+Sinus+Sph*)** achieves the lowest `mach` RRMSE (**0.01558**), improving over the strengthened **MGN** (**0.03099**) and **MGN Tree** (**0.01908**) by **49.7%** and **18.3%**, respectively. On `nut`, all (15/80) configurations cluster tightly around **0.152–0.154**. Overall, these strengthened comparisons confirm that **MiSe-GNN**'s advantage on `mach` persists even when baselines receive identical geometric encodings, supporting the claim that selective, error-driven augmentation not feature engineering alone drives the observed improvements.

**Comparison to PLAID benchmarks:** Against public PLAID baselines (Table 16), MARIO leads with *Total Error* = **0.01855**. **MiSe-GNN** ranks #4 overall but is second best on `mach` (0.0156), highlighting competitive performance on the speed field, while `nut` remains the main bottleneck.

*Table 16.* `VKI-LS59` PLAID benchmarks (`nut` and `mach` fields only). (*) MiSe-GNN (15/80) results represent the best performance across all tested feature engineering configurations. **Bold** indicates best, underline indicates second best. Results obtained on 14/10/2025.

| Rank | ID | Total Error | nut | mach |
|---|---|---|---|---|
| **1** | **MARIO** | **0.01855** | **0.0259** | **0.0112** |
| 2 | Augur | 0.03225 | 0.0424 | 0.0221 |
| 3 | Vi-Transformer | 0.03650 | 0.0498 | 0.0232 |
| 4 | MiSe-GNN[*] | 0.04480 | 0.0740 | 0.0156 |
| 5 | FNO | 0.05130 | 0.0846 | 0.0180 |
| 6 | MMGP+ | 0.05655 | 0.0822 | 0.0309 |

### D.2. `Tensile2d` dataset

**Base setting (10 MP / hidden size 32):** Table 17 shows that **MiSe-GNN** dramatically outperforms **MGN** on all five fields (reductions of **64%–95%** depending on the field), and improves over **MGN Tree** by **9%–11%** on `U1`, `U2`, `sig11`, and `sig22`, on `sig12` the two methods are essentially on par. Macro-averaged across all five fields, **MiSe-GNN** (10/32) reaches (**0.00481**), lower than **MGN Tree** (**0.00527**). See Figure 23 for qualitative comparisons on field `U1`.

*Table 17.* Results on `Tensile2d` (10 MP / hidden size 32).

| Model | U1 | U2 | sig11 | sig22 | sig12 |
|---|---|---|---|---|---|
| MGN | 0.09236 | 0.05658 | 0.16486 | 0.01315 | 0.00789 |
| MGN Tree | 0.00550 | 0.00745 | 0.00836 | 0.00231 | **0.00274** |
| **MiSe-GNN** | **0.00499** | **0.00663** | **0.00760** | **0.00207** | 0.00278 |
| $\Delta_{\text{MGN}}$ | 94.60% | 88.28% | 95.39% | 84.26% | 64.77% |
| $\Delta_{\text{MGN Tree}}$ | 9.27% | 11.01% | 9.09% | 10.39% | -1.46% |

**Capacity scaling & feature engineering (15 MP / hidden size 80):** Table 18 shows that *Sinus+Sph* yields the best macro-average RRMSE (**0.00305**, **−36.6%** vs. MiSe-GNN 10/32) and achieves the lowest errors on the stress components

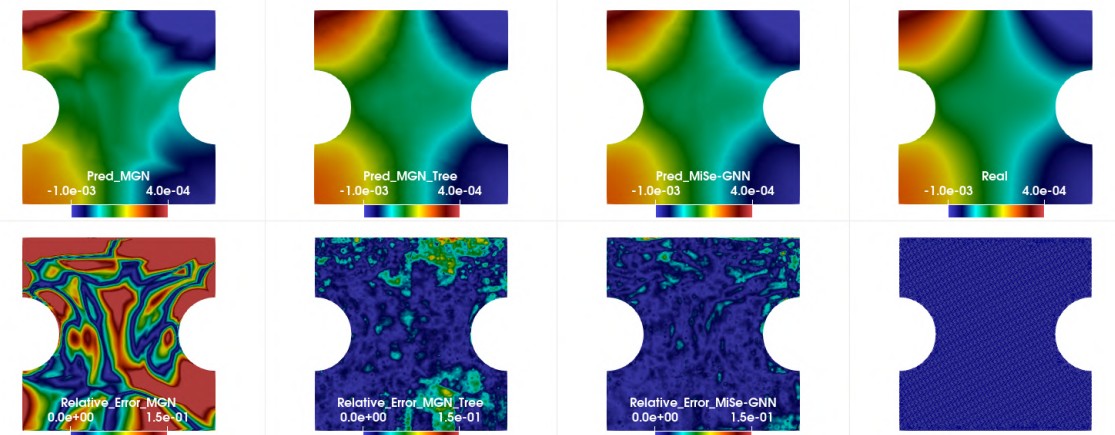

*Figure 23.* Qualitative comparison on `Tensile2d`. Top row: predicted fields, bottom row: relative errors. From left to right: **MGN**, **MGN Tree**, **MiSe-GNN**, and ground truth.

(`sig11`,`sig22`,`sig12`). OurFeat primarily benefits the displacement fields: **OurFeat+*Sinus+Sph*** attains the best `U1` and `U2`, while slightly degrading stress accuracy, resulting in a slightly higher macro-average (**0.00316**). Overall, multi-frequency encodings dominate stress prediction, while **OurFeat** provides complementary geometry cues for displacements.

*Table 18.* Results on `Tensile2d` with capacity scaling and feature engineering (15 MP / hidden size 80).

| Model | U1 | U2 | sig11 | sig22 | sig12 |
|---|---|---|---|---|---|
| MiSe-GNN | 0.00402 | 0.00609 | 0.00597 | 0.00178 | 0.00226 |
| **MiSe-GNN (Sinus+Sph)** | 0.00343 | 0.00428 | **0.00466** | **0.00133** | **0.00157** |
| MiSe-GNN (OurFeat) | 0.00328 | 0.00432 | 0.00523 | 0.00159 | 0.00219 |
| MiSe-GNN (OurFeat+Sinus+Sph) | **0.00289** | **0.00402** | 0.00497 | 0.00160 | 0.00231 |
| MGN (Sinus+Sph) | 0.02520 | 0.02351 | 0.07957 | 0.00929 | 0.00366 |
| MGN Tree (Sinus+Sph) | 0.00462 | 0.00594 | 0.00597 | 0.00144 | 0.00151 |

We also run **MGN** and **MGN Tree** with the same *Sinus+Sph* features under the (15 MP / hidden size 80) setting. Table 18 shows that **MiSe-GNN (*Sinus+Sph*)** achieves a macro-average RRMSE of **0.00305**, compared to **0.02825** for **MGN (*Sinus+Sph*)** and **0.00390** for **MGN Tree (*Sinus+Sph*)**, corresponding to reductions of **89.2%** and **21.6%**, respectively. Per-field, MiSe-GNN reduces errors relative to the strengthened **MGN Tree** by **25–28%** on displacements (`U1`/`U2`) and by **8–22%** on key stresses (`sig11`/`sig22`), while remaining highly competitive on `sig12` (**0.00157** vs. **0.00151**). These strengthened comparisons indicate that, beyond harmonic feature encodings and hierarchical rewiring, **MiSe-GNN**'s error-guided connectivity provides additional gains, especially for propagating long-range mechanical interactions relevant to displacement and stress recovery.

*Table 19.* `Tensile2d` PLAID benchmarks (`U1`, `U2`, `sig11`, `sig22` and `sig12` fields only). (*) MiSe-GNN (15/80) results represent the best performance across all tested feature engineering configurations. **Bold** indicates best, underline indicates second best. Results obtained on 14/10/2025.

| Rank | ID | Total Error | U1 | U2 | sig11 | sig22 | sig12 |
|---|---|---|---|---|---|---|---|
| **1** | **MMGP** | **0.00178** | **0.0015** | **0.0009** | **0.0031** | **0.0013** | 0.0021 |
| 2 | MARIO | 0.00266 | 0.0023 | 0.0030 | 0.0040 | 0.0017 | 0.0023 |
| 3 | MiSe-GNN[*] | 0.00288 | 0.0029 | 0.0040 | 0.0047 | **0.0013** | **0.0016** |
| 4 | Augur | 0.00502 | 0.0037 | 0.0048 | 0.0081 | 0.0035 | 0.0050 |
| 5 | MMVT | 0.00542 | 0.0043 | 0.0051 | 0.0089 | 0.0037 | 0.0051 |
| 6 | Vi-Transformer | 0.01218 | 0.0086 | 0.0091 | 0.0184 | 0.0102 | 0.0146 |
| 7 | FNO | 0.01452 | 0.0174 | 0.0110 | 0.0250 | 0.0057 | 0.0135 |

**Comparison to PLAID benchmarks:** On the PLAID leaderboard (Table 19), **MiSe-GNN** places #3 overall with *Total Error* (**0.00288**), while ranking **first** on `sig12` and tying for **first** on `sig22`. MMGP and MARIO lead overall, but **MiSe-GNN** is especially strong on stress components.

## E. Analysis & Ablations

### E.1. Budget Analysis

We evaluate training under two realistic budgets on `2D_profile` dataset: (i) an *epoch-limited* regime that fixes the number of optimization steps, and (ii) a *time-limited* regime that fixes the wall-clock training time. For fairness, we train *one model per field* and report the *macro-average RRMSE* over the fields in all plots and tables. All methods are trained and timed on the same hardware and dataloading. In MiSe-GNN, the per-epoch cost is higher due to (a) additional edges are dynamically constructed for graph augmentation, increasing the message-passing workload and (b) the model jointly optimizes two tasks predicting both *fields* and their corresponding *errors*. The question is whether that extra cost buys better accuracy *per compute*.

**Epoch-limited results:** Figure 24 shows mean RRMSE versus training epochs, with markers at {100, 200, 500}. Despite the higher per-epoch compute, MiSe-GNN attains substantially lower error than both MGN and MGN Tree at the same epoch budgets, and the gap widens with more training [3]. Table 20 summarizes the key numbers.

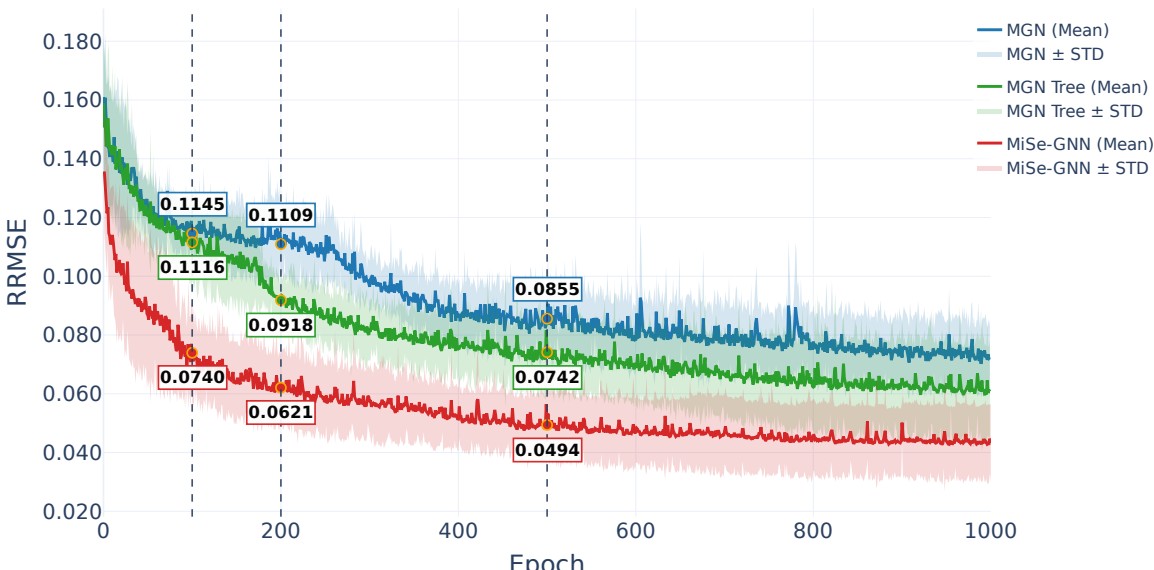

*Figure 24.* RRMSE vs. epoch (markers at 100/200/500 epoch).

*Table 20.* Epoch-limited summary: RRMSE (mean $\pm$ std) and percentage improvement ($\Delta$).

| Model | @100 epoch | @200 epoch | @500 epoch |
|---|---|---|---|
| MGN | $0.1145 \pm 0.0102$ | $0.1109 \pm 0.0124$ | $0.0855 \pm 0.0079$ |
| MGN Tree | $0.1116 \pm 0.0127$ | $0.0918 \pm 0.0085$ | $0.0742 \pm 0.0140$ |
| **MiSe-GNN** | $\mathbf{0.0740 \pm 0.0167}$ | $\mathbf{0.0621 \pm 0.0125}$ | $\mathbf{0.0494 \pm 0.0110}$ |
| $\Delta_{\text{MGN}}$ | 35.37% | 44.00% | 42.22% |
| $\Delta_{\text{MGN Tree}}$ | 33.69% | 32.35% | 33.42% |

---

[3]MGN Tree can be competitive on `Pressure`, but this does not persist across the other fields, the macro-average therefore favors MiSe-GNN.

**Time-limited results:** Although MiSe-GNN trains more slowly *per epoch*, the *anytime* curve in Figure 25 shows that it reaches useful accuracy *earlier in wall-clock time*. We quantify this in two complementary ways.

*(i) Time-to-target:* Table 21 reports the minutes each method needs to match the accuracy that MGN attains at 100/200/500 epochs, MiSe-GNN is 3–4× faster at early milestones. Table 22 inverts the target and shows that baselines fail to reach MiSe-GNN's 200/500-epoch accuracy within the logged time.

*(ii) Equal-time anchors:* We also compare errors at fixed timestamps corresponding to when MGN reaches 100/200/500 epochs (Table 23). Under these equal-time budgets, MiSe-GNN achieves the lowest mean RRMSE at all anchors.

Overall, MiSe-GNN is Pareto-efficient in the accuracy–time trade-off: the extra per-epoch cost from edge augmentation is amortized by faster error reduction per unit time.

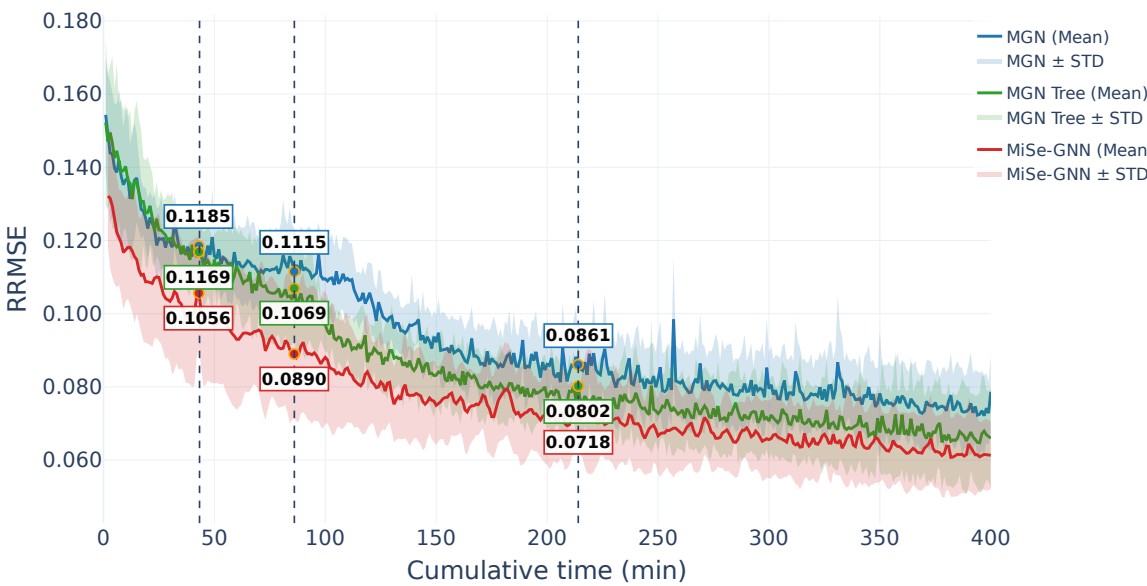

*Figure 25.* **Time-limited training.** Although MiSe-GNN has a higher per-epoch cost (edge augmentation + multi-task), it achieves substantially lower error for the same budget and traces a better *anytime* curve.

*Table 21.* Wall-clock minutes to reach MGN's accuracy. MiSe-GNN is 3–4× faster at early milestones.

| Target Epoch | Target Model | Target RRMSE | Time to Target (min) | | |
|---|---|---|---|---|---|
| | | | MGN | MGN Tree | **MiSe-GNN** |
| 100 | MGN | 0.1145 | 44.74 | 44.62 | **15.08** |
| 200 | MGN | 0.1109 | 69.86 | 48.91 | **16.68** |
| 500 | MGN | 0.0855 | 177.90 | 136.89 | **101.84** |

**Takeaways:** (i) Under the same *epoch* budget, MiSe-GNN dominates the accuracy frontier (macro-average across fields). (ii) Under a fixed *time* budget, targeting persistent high-error regions leads to faster error reduction, MiSe-GNN reaches a given accuracy in markedly less time. (iii) The higher per-epoch cost of augmentation is amortized by improved convergence *per unit time*.

*Table 22.* Minutes needed to reach MiSe-GNN's accuracy. Baselines fail to reach the 200/500 epoch targets. "—" denotes target not reached.

| Target Epoch | Target Model | Target RRMSE | Time to Target (min) | | |
|---|---|---|---|---|---|
| | | | MGN | MGN Tree | **MiSe-GNN** |
| 100 | MiSe-GNN | 0.0740 | 367.40 | 244.24 | **162.90** |
| 200 | MiSe-GNN | 0.0621 | — | 469.67 | **361.54** |
| 500 | MiSe-GNN | 0.0494 | — | — | **702.81** |

*Table 23.* Equal-time summary table: Comparison of RRMSE and relative gains under fixed wall-clock time budgets.

| Baseline Time (min) | RRMSE (Mean $\pm$ Std) | | | Gain vs. MGN | |
|---|---|---|---|---|---|
| | MGN | MGN Tree | **MiSe-GNN** | Tree | **MiSe** |
| 43.3 | $0.1185 \pm 0.0089$ | $0.1169 \pm 0.0109$ | $\mathbf{0.1056 \pm 0.0150}$ | 1.35% | **10.89%** |
| 86.1 | $0.1115 \pm 0.0118$ | $0.1069 \pm 0.0128$ | $\mathbf{0.0890 \pm 0.0177}$ | 4.13% | **20.18%** |
| 214.0 | $0.0861 \pm 0.0086$ | $0.0802 \pm 0.0145$ | $\mathbf{0.0718 \pm 0.0083}$ | 6.85% | **16.61%** |

## E.2. Ablation Study

We conduct a systematic ablation study on the MiSe-GNN framework to analyze the sensitivity of model performance with respect to key hyperparameters involved in the adaptive graph augmentation procedure. Specifically, we evaluate the impact of:

- **Augmentation Period ($T$ epochs):** The frequency at which error-guided edges are updated.

- **Error Threshold ($\tau_{\mathrm{err}}$):** The cutoff value for selecting high-error nodes to augment.

- **Ancestor Hops ($k$):** The number of ancestor hops considered in the augmentation process.

Additionally, we investigate the impact of the **error loss weight** ($\lambda_{\mathrm{err}}$), which regulates the strength of error supervision during training.

The experiments are performed on the challenging u1 field of the 2D_MultiScHypEl dataset. We report the Relative Root Mean Square Error (RRMSE), edge augmentation time, and total training time as key evaluation metrics. Default configuration: $T = 10$, $\tau_{\mathrm{err}} = 0.02$, $k = 3$, and $\lambda_{\mathrm{err}} = 1$ unless otherwise noted.

### E.2.1. IMPACT OF AUGMENTATION PERIOD ($T$)

Table 24 shows that more frequent augmentation improves accuracy but increases augmentation overhead. At $T$=5, RRMSE is lowest (0.01429) with the highest augmentation time, $T$=10 offers a strong accuracy–cost balance (RRMSE 0.01464), while $T$=20 reduces augmentation overhead further at a notable accuracy cost (RRMSE 0.01608). Total training time varies modestly across settings on our hardware (1 GPU A30), we adopt $T$=10 by default.

*Table 24.* Effect of augmentation period $T$ on MiSe-GNN performance.

| T | RRMSE | Edge Aug Time (s) | Total Training Time (s) |
|---|---|---|---|
| 5 | 0.01429 | 112714 | 124708 |
| **10** | **0.01464** | **49173** | **125371** |
| 20 | 0.01608 | 26737 | 127845 |

### E.2.2. IMPACT OF ERROR THRESHOLD ($\tau_{\mathrm{ERR}}$)

A moderate threshold $\tau_{\mathrm{err}}{=}0.02$ yields the best trade-off (Table 25): it selects sufficiently many high-error nodes to be informative without introducing redundant edges. Lowering $\tau$ to 0.01 slightly increases compute but does not improve accuracy, raising it to 0.05 degrades accuracy.

*Table 25.* Effect of error threshold $\tau$ on MiSe-GNN performance.

| $\tau$ | RRMSE | Edge Aug Time (s) | Total Training Time (s) |
|------|---------|-------|--------|
| 0.01 | 0.01473 | 47660 | 127812 |
| **0.02** | **0.01464** | **49173** | **125371** |
| 0.05 | 0.01558 | 49405 | 124441 |

### E.2.3. IMPACT OF ANCESTOR HOPS ($k$)

Table 26 compares MiSe-GNN performance at varying ancestor hops ($k$) with 15 message-passing steps. Moderate ancestor hops ($k = 3, 5$) yield similar accuracy, with $k{=}5$ slightly outperforming at marginal overhead. We thus select $k{=}3$ as a simpler, robust default. Results with 10 message-passing steps (Table 27) confirm this trend.

*Table 26.* Effect of ancestor hops $k$ on MiSe-GNN performance (15 MP).

| $k$ | RRMSE | Edge Aug Time (s) | Total Training Time (s) |
|---|---------|-------|--------|
| 1 | 0.01491 | 52744 | 129985 |
| **3** | **0.01464** | **49173** | **125371** |
| 5 | 0.01451 | 49365 | 123789 |

*Table 27.* Effect of ancestor hops $k$ on MiSe-GNN performance (10 MP).

| $k$ | RRMSE | Edge Aug Time (s) | Total Training Time (s) |
|---|---------|-------|--------|
| 1 | 0.01834 | 51305 | 89422 |
| **3** | **0.01665** | **47640** | **87091** |
| 5 | 0.01790 | 46248 | 85640 |

### E.2.4. IMPACT OF ERROR LOSS WEIGHT ($\lambda_{\mathrm{err}}$)

Table 28 indicates that $\lambda_{\mathrm{err}}{=}1$ achieves the best overall accuracy; both under and over weighting the error head (0.25 or 4) slightly hurt RRMSE and increase variance in training time.

*Table 28.* Effect of the error loss weight $\lambda_{\mathrm{err}}$ on MiSe-GNN performance.

| $\lambda_{\mathrm{err}}$ | RRMSE | Edge Aug Time (s) | Total Training Time (s) |
|------|---------|-------|--------|
| 0.25 | 0.01622 | 51262 | 126692 |
| **1** | **0.01464** | **49173** | **125371** |
| 4 | 0.01508 | 52557 | 127204 |

E.2.5. SUMMARY OF ABLATION FINDINGS

Our experiments reveal clear trade-offs in hyperparameter selection:

- Frequent augmentation enhances accuracy but increases computational overhead.

- Moderate error thresholds effectively identify informative edges without excessive redundancy.

- Intermediate ancestor hop values ($k$) achieve efficient multiscale information propagation.

- Balanced error loss weighting ensures stable error prediction and optimal overall performance.

These findings offer clear defaults for practical tuning, demonstrating that MiSe-GNN's performance derives from targeted, error-aware augmentation rather than indiscriminate graph densification. This confirms the method's robustness and provides practical guidelines for hyperparameter tuning in graph-based PDE prediction tasks.

**Reproducibility:**   All ablations were conducted consistently using identical hardware/software configurations and training protocols. Timings reported (seconds) in Tables 24–28 enable accurate reproduction of results.

### E.3. Edge Refresh, Pruning, and Base Mesh Retention

Because $\mathcal{E}^\star$ is rebuilt from the current predicted error map at every augmentation update, the number of augmentation edges can decrease as the model improves. Table 29 shows one representative 2D_profile training run, where the number of augmentation edges decreases over training as fewer nodes remain above the error threshold.

*Table 29.* Number of augmentation edges during training for one representative 2D_profile sample.

| Epoch | Number of augmentation edges |
|-------|------------------------------|
| 10    | 72692                        |
| 500   | 33567                        |
| 1000  | 17484                        |

We intentionally retain the original mesh edges $\mathcal{E}_b$ because they encode local discretization couplings of the underlying PDE. Removing them and using only tree induced edges degrades performance, as shown in Table 30. Thus MiSe-GNN preserves reliable local physical connectivity while adaptively adding augmentation pathways only where needed.

*Table 30.* Effect of retaining the original mesh graph on the u1 field of 2D_MultiScHypEl.

| Model | Test RRMSE |
|-------|------------|
| MiSe-GNN (tree-only base graph, original mesh removed) | 0.0243 |
| MiSe-GNN (full base graph, original mesh retained) | **0.0185** |

### E.4. Equal-Budget Edge Allocation Controls

To isolate whether the improvement of MiSe-GNN comes from error-guided allocation rather than from simply increasing the number of edges, we compare against non-error-guided augmentation policies with matched average augmentation-edge budgets. The controls include random augmentation edges, distance-based augmentation edges, and static hierarchical tree edges. Table 31 reports macro-average RRMSE values, while Tables 32-36 provide per-field results. Relative reductions are computed as:

$$\Delta = 100 \times \frac{\mathrm{RRMSE_{control}} - \mathrm{RRMSE_{MiSe}}}{\mathrm{RRMSE_{control}}}.$$

Because table entries are rounded for readability, the reported percentages are computed from unrounded per-field values.

Figures 26-27 qualitatively illustrate the same conclusion: random and distance-based policies match the augmentation-edge budget but do not adapt the allocation to the learned error structure.

*Table 31.* Equal-budget augmentation controls. Values are macro-average RRMSE; lower is better. Parenthesized values indicate the relative RRMSE reduction achieved by MiSe-GNN over each corresponding control under the formula above.

| Model | Augmentation-edge policy | 2D_profile | 2D_profile Uniform | 2D_MultiScHypEl | VKI-LS59 | Tensile2d |
|---|---|---|---|---|---|---|
| MGN | none | 0.0356 (+17.0%) | 0.0274 (+24.5%) | 0.0269 (+16.6%) | 0.0310 (+49.7%) | 0.0282 (+89.2%) |
| MGN Tree | static hierarchical | 0.0376 (+21.4%) | 0.0250 (+17.3%) | 0.0261 (+14.1%) | 0.0191 (+18.3%) | 0.0039 (+21.6%) |
| MGN Random | random | 0.0383 (+22.9%) | 0.0251 (+17.4%) | 0.0242 (+7.4%) | 0.0244 (+36.1%) | 0.0039 (+21.5%) |
| MGN Dist-based | distance-based | 0.0368 (+19.7%) | 0.0266 (+22.3%) | 0.0251 (+10.9%) | 0.0307 (+49.3%) | 0.0256 (+88.1%) |
| MiSe-GNN | error-guided | 0.0295 | 0.0207 | 0.0224 | 0.0156 | 0.0031 |

*Table 32.* Per-field equal-budget results on 2D_profile. Values are RRMSE; lower is better.

| Model | Mach | Pressure | Velocity-x | Velocity-y | Macro |
|---|---|---|---|---|---|
| MGN | 0.03869 | 0.02746 | 0.04600 | 0.03018 | 0.0356 |
| MGN Tree | 0.04147 | 0.02838 | 0.04709 | 0.03328 | 0.0376 |
| MGN Random | 0.04330 | 0.03000 | 0.04777 | 0.03212 | 0.0383 |
| MGN Dist-based | 0.04034 | 0.02836 | 0.04684 | 0.03161 | 0.0368 |
| MiSe-GNN | **0.03088** | **0.02543** | **0.03576** | **0.02603** | **0.0295** |

*Table 33.* Per-field equal-budget results on the uniform-mesh version of 2D_profile. Values are RRMSE; lower is better.

| Model | Mach | Pressure | Velocity-x | Velocity-y | Macro |
|---|---|---|---|---|---|
| MGN | 0.02711 | 0.02431 | 0.02744 | 0.03086 | 0.0274 |
| MGN Tree | 0.02607 | 0.01887 | 0.02543 | 0.02978 | 0.0250 |
| MGN Random | 0.02528 | 0.02265 | 0.02530 | 0.02700 | 0.0251 |
| MGN Dist-based | 0.02662 | 0.02400 | 0.02693 | 0.02903 | 0.0266 |
| MiSe-GNN | **0.02129** | **0.01660** | **0.02037** | **0.02457** | **0.0207** |

*Table 34.* Per-field equal-budget results on 2D_MultiScHypEl. Values are RRMSE; lower is better.

| Model | u1 | u2 | P11 | P12 | P22 | P21 | psi | Macro |
|---|---|---|---|---|---|---|---|---|
| MGN | 0.01638 | 0.01761 | 0.02258 | 0.03779 | 0.02318 | 0.03635 | 0.03420 | 0.0269 |
| MGN Tree | 0.01375 | 0.01597 | 0.02165 | 0.03605 | 0.02206 | 0.03881 | 0.03436 | 0.0261 |
| MGN Random | 0.01407 | 0.01483 | 0.01950 | 0.03345 | 0.01990 | 0.03354 | 0.03409 | 0.0242 |
| MGN Dist-based | 0.01527 | 0.01639 | 0.01994 | 0.03629 | 0.01970 | 0.03465 | 0.03377 | 0.0251 |
| MiSe-GNN | **0.01203** | **0.01450** | **0.01730** | **0.03119** | **0.01764** | **0.03140** | **0.03282** | **0.0224** |

*Table 35.* Equal-budget results on VKI-LS59 mach prediction. Values are RRMSE; lower is better.

| Model | Policy | RRMSE |
|---|---|---|
| MGN | none | 0.03099 |
| MGN Tree | static hierarchical | 0.01908 |
| MGN Random | random | 0.02438 |
| MGN Dist-based | distance-based | 0.03073 |
| MiSe-GNN | error-guided | **0.01558** |

*Table 36.* Per-field equal-budget results on Tensile2d. Values are RRMSE; lower is better. MiSe-GNN achieves the best macro RRMSE, while random edges are slightly better on two stress components.

| Model | U1 | U2 | sig11 | sig22 | sig12 | Macro |
|---|---|---|---|---|---|---|
| MGN | 0.02520 | 0.02351 | 0.07957 | 0.00929 | 0.00366 | 0.0282 |
| MGN Tree | 0.00462 | 0.00594 | 0.00597 | 0.00144 | 0.00151 | 0.0039 |
| MGN Random | 0.00369 | 0.00629 | 0.00674 | **0.00124** | **0.00148** | 0.0039 |
| MGN Dist-based | 0.01784 | 0.01511 | 0.08502 | 0.00699 | 0.00288 | 0.0256 |
| MiSe-GNN | **0.00343** | **0.00428** | **0.00466** | 0.00133 | 0.00157 | **0.0031** |

## E.5. Alternative Training Variants

MiSe-GNN supervises the error head on the base graph, rather than on the changing augmented graph. This is intentional: the error head learns a stable pre-intervention diagnostic signal indicating where the base graph fails. To test this design choice, we compare with an iterative variant that predicts error on the current augmented graph and then rebuilds the graph from that prediction. Although this variant can train faster, it produces less accurate solutions, as shown in Table 37.

We also compare against a two-model pipeline in which one model predicts the error on the base graph and a second model

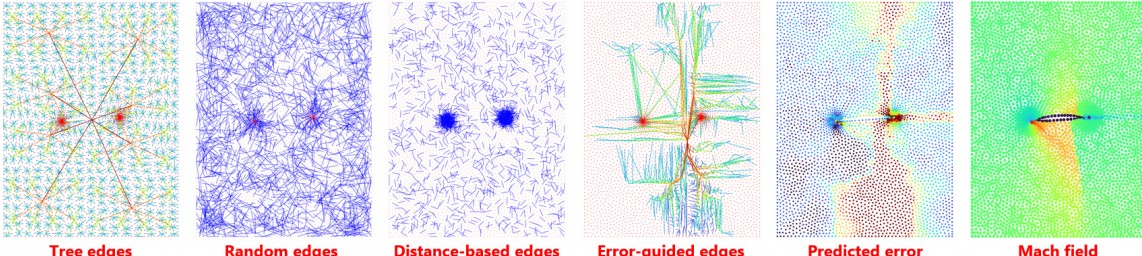

*Figure 26.* Qualitative comparison of augmentation edge policies on the uniform-mesh `2D_profile` case. Static tree, random, and distance-based policies allocate edges independently of the learned error structure, whereas MiSe-GNN concentrates augmentation connectivity around regions of high predicted error and strong field variation.

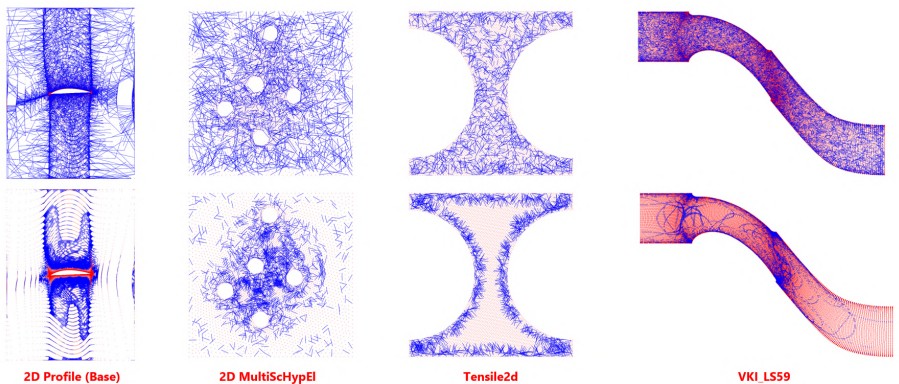

*Figure 27.* Qualitative examples of random and distance-based equal-budget augmentation edges across datasets. Top row: random edges. Bottom row: distance-based edges. These policies match the augmentation-edge budget but do not adapt the allocation to the predicted error structure.

*Table 37.* Base-graph error supervision versus iterative augmented-graph error supervision.

| Dataset | Target | Test RRMSE |
|---|---|---|
| `2D_MultiScHypEl` | `u1`, MiSe-GNN | **0.0185** |
| `2D_MultiScHypEl` | `u1`, iterative variant | 0.0234 |
| `2D_profile` | `Pressure`, MiSe-GNN | **0.0290** |
| `2D_profile` | `Pressure`, iterative variant | 0.0333 |

predicts the field on the augmented graph:

$$G_b \rightarrow \text{Error model} \rightarrow \hat{\boldsymbol{\epsilon}} \rightarrow G_a \rightarrow \text{Field model} \rightarrow \hat{\mathbf{y}}.$$

On the `u1` task of `2D_MultiScHypEl`, this two-model variant improves over MGN and MGN Tree but remains below the shared dual-head MiSe-GNN, as shown in Table 38. This suggests that the gains come not only from graph refinement, but also from joint representation learning between error prediction and field prediction.

*Table 38.* Two-model pipeline compared with the shared dual-head MiSe-GNN on `2D_MultiScHypEl u1`.

| Model | Test RRMSE |
|---|---|
| MGN | 0.0419 |
| MGN Tree | 0.0249 |
| Multi-model GNN | 0.0221 |
| MiSe-GNN | **0.0185** |

## E.6. Adaptive Tree Construction Cost

Table 39 reports the average time required to build error-guided augmentation edges per sample across the evaluated datasets. Although MiSe-GNN has a higher per-epoch cost than the base GNN, the accuracy-time analysis in Appendix E.1 shows that the additional cost is amortized by faster convergence and better anytime performance.

*Table 39.* Average time to build error-guided augmentation edges per sample.

| Dataset | Number of samples | Time / sample |
|---|---|---|
| 2D_MultiScHypEl | 1140 | 0.435 s |
| 2D_profile | 400 | 0.519 s |
| VKI-LS59 | 839 | 0.579 s |
| Tensile2d | 700 | 0.167 s |

# F. Error Graph

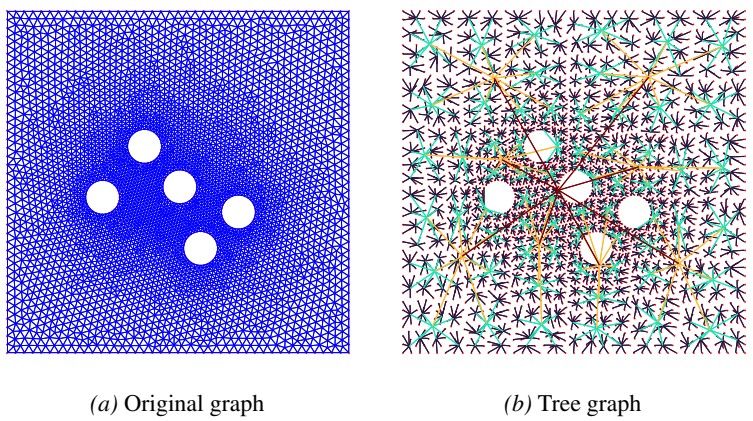

*(a)* Original graph      *(b)* Tree graph

*Figure 28.* Original mesh and corresponding tree generated by MGN Tree (2D_MultiScHypEl).

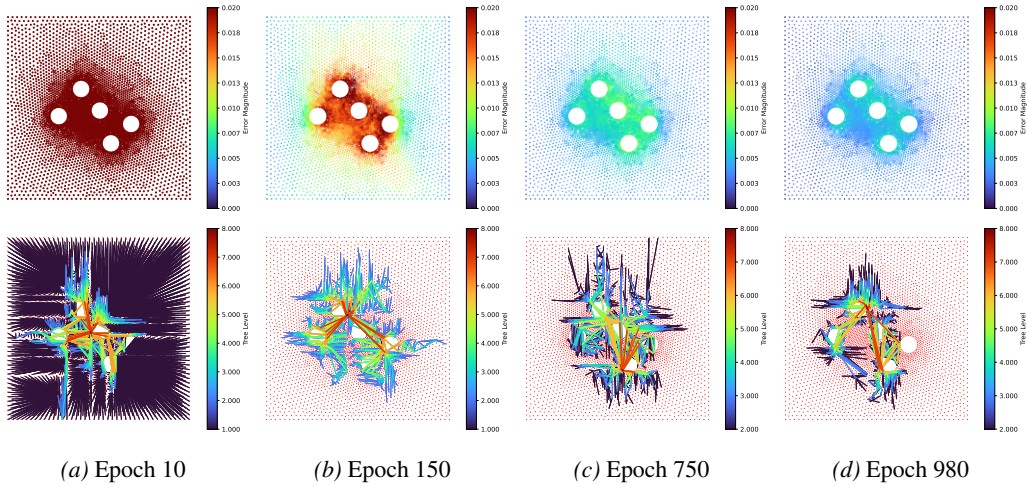

*(a)* Epoch 10      *(b)* Epoch 150      *(c)* Epoch 750      *(d)* Epoch 980

*Figure 29.* Evolution of the augmented graph during training. Top row: predicted errors ($\hat{\epsilon}_{base}^{(t)}$), bottom row: set of error-guided edges ($\mathcal{E}^{\star}$).

## G. Error Head Diagnostic Fidelity

### G.1. Motivation and Metric Definitions

To evaluate the capability of the error head in identifying physical complexities, we assess the alignment between the predicted error field $\hat{\boldsymbol{\epsilon}}_{base} \in \mathbb{R}^N$ and the ground-truth relative error $\boldsymbol{\epsilon}^\star \in \mathbb{R}^N$, where $N$ is the number of nodes. We employ three complementary metrics:

**1. Spearman Rank Correlation ($r_s$):** This metric assesses whether the model correctly ranks nodes from low to high error, independent of the absolute scale. It is computed as the Pearson correlation coefficient between the rank variables:

$$r_s = \frac{\text{cov}(\text{rg}(\hat{\boldsymbol{\epsilon}}_{base}), \text{rg}(\boldsymbol{\epsilon}^\star))}{\sigma_{\text{rg}(\hat{\boldsymbol{\epsilon}}_{base})}\sigma_{\text{rg}(\boldsymbol{\epsilon}^\star)}} \tag{21}$$

where $\text{rg}(\mathbf{v})$ denotes the vector of ranks for the elements in $\mathbf{v}$. A high $r_s$ ensures that the rewiring mechanism prioritizes nodes in the correct order of difficulty.

**2. Top-$k$ Recall ($R_k$):** Let $\mathcal{I}_{\text{pred}}^k$ be the set of indices corresponding to the $k$ nodes with the highest predicted errors, and $\mathcal{I}_{\text{true}}^k$ be the equivalent set for the ground truth errors, where $k = \lfloor \alpha N \rfloor$ (e.g., $\alpha = 0.05$ for top-5%). Recall measures the intersection of these sets:

$$R_k = \frac{|\mathcal{I}_{\text{pred}}^k \cap \mathcal{I}_{\text{true}}^k|}{k} \tag{22}$$

This metric quantifies the model's success in spatially localizing the most critical regions (e.g., shock fronts).

**3. Top-$k$ Error Mass Capture ($M_k$):** Standard recall treats all top-$k$ nodes equally. However, in PDE problems, a small subset of nodes often contributes disproportionately to the total error. We propose *Error Mass Capture* to measure the fraction of the *oracle* top-$k$ ground-truth error mass recovered by the predicted top-$k$ nodes:

$$M_k = \frac{\sum_{i \in \mathcal{I}_{\text{pred}}^k} \boldsymbol{\epsilon}_i^\star}{\sum_{j \in \mathcal{I}_{\text{true}}^k} \boldsymbol{\epsilon}_j^\star} \tag{23}$$

Here, the denominator represents the maximum possible error mass that an ideal oracle could capture with $k$ nodes. A value of $M_k \approx 1$ implies that the rewiring strategy, guided by $\hat{\boldsymbol{\epsilon}}_{base}$, effectively targets the dominant sources of simulation error. $M_k$ is particularly relevant for mesh-based PDEs where errors are often concentrated in small, high-gradient regions.

### G.2. Reporting (mean over test samples)

Let the test set contain $M$ samples. We compute $\{r_s^{(m)}, R_k^{(m)}, M_k^{(m)}\}_{m=1}^M$ and report the mean over samples:

$$\overline{r}_s = \frac{1}{M} \sum_{m=1}^M r_s^{(m)}, \quad \overline{R}_k = \frac{1}{M} \sum_{m=1}^M R_k^{(m)}, \quad \overline{M}_k = \frac{1}{M} \sum_{m=1}^M M_k^{(m)} \tag{24}$$

To isolate the intrinsic diagnostic capability of the architecture, we evaluate a standardized model configuration (15 MP / hidden size 80) across all four benchmarks, without relying on any auxiliary feature engineering. We apply the proposed metrics to all physical fields; the aggregate results are reported in Table 40.

**Analysis of Diagnostic Fidelity:** Table 40 demonstrates that the error head provides a meaningful ranking/localization signal across diverse physics regimes, ranging from compressible flows in `2D_profile` to nonlinear mechanics in `Tensile2d`. Averaged over all 18 field targets, we obtain a mean Spearman correlation of $r_s = 0.476$ and a Top-5% recall of $R_{5\%} = 0.421$ (range 0.143–0.763), indicating moderate but consistent localization of the hardest regions. More importantly for driving augmentation, the Top-5% error mass capture averages $M_{5\%} = 0.666$ and reaches up to 0.903 on

*Table 40.* Diagnostic fidelity of the Error Head across all benchmarks. Metrics are computed on the test set (Top-5% for Recall and Mass Capture). Mass Capture ($M_{5\%}$) reports the fraction of the *oracle* Top-5% ground-truth error mass captured by the nodes selected by the predicted Top-5% set (i.e., nodes prioritized for graph augmentation).

| Dataset | Physical Field | Spearman $r_s$ | Recall ($R_{5\%}$) | Mass Capture ($M_{5\%}$) |
|---|---|---|---|---|
| 2D_MultiScHypEl | u1 | 0.4620 | 0.1427 | 0.4652 |
| | u2 | 0.4949 | 0.1460 | 0.4737 |
| | P11 | 0.3674 | 0.3578 | 0.6357 |
| | P12 | 0.3122 | 0.3393 | 0.6134 |
| | P22 | 0.3713 | 0.3402 | 0.6397 |
| | P21 | 0.3312 | 0.3402 | 0.6098 |
| | psi | **0.6486** | **0.4674** | **0.6853** |
| 2D_profile | Mach | 0.6461 | 0.5065 | 0.7203 |
| | Pressure | 0.4718 | 0.4574 | 0.6737 |
| | Velocity-x | 0.6468 | 0.5205 | 0.7281 |
| | Velocity-y | **0.7097** | **0.7633** | **0.8908** |
| VKI-LS59 | nut | **0.6280** | 0.6133 | 0.7957 |
| | mach | 0.5392 | **0.7634** | **0.9028** |
| Tensile2d | U1 | 0.3083 | 0.3710 | 0.6375 |
| | U2 | **0.4631** | 0.3588 | 0.6221 |
| | sig11 | 0.3689 | **0.3940** | 0.6369 |
| | sig22 | 0.3639 | 0.3267 | 0.6094 |
| | sig12 | 0.4278 | 0.3778 | **0.6542** |

shock-dominated `mach` fields, meaning that the nodes selected by the predicted Top-5% set recover a large fraction of the dominant error contributors. The gap between $R_{5\%}$ and $M_{5\%}$ suggests a value-weighted behavior: even when the predicted top-k set does not perfectly match the oracle set, it tends to focus on the highest-mass error regions that matter most for reducing global error.

This discrepancy highlights a crucial property of **MiSe-GNN**: the error head acts as a *value-based attention mechanism*. Rather than broadly flagging all nodes with non-zero errors (which would maximize Recall but dilute computational resources), the model successfully locks onto the heavy hitters, shocks, and high-gradient regions that dominate the global error norm. Crucially, this high diagnostic fidelity is achieved using raw features alone, confirming that the dual-head architecture learns to identify physical complexity end-to-end, validating its role as a robust driver for adaptive graph augmentation.

