# OpenReview forum: "Error-Driven Graph Augmentation for Mesh-Based PDE Surrogates"
_ICML.cc/2026/Conference — ICML 2026 regular_

### Official Review · Reviewer_C3UB · 2026-03-08

**Soundness:** 3
**Presentation:** 4
**Significance:** 3
**Originality:** 3
**Overall Recommendation:** 4
**Confidence:** 4

**Summary:**

This work proposed using a posteriori error estimation for adaptive graph augmentation, in order to overcome the limitations of the finite reception area of message passing graph networks. The field prediction and the error prediction are separately trained with two decoder heads. The augmented edges are updated every T epochs. Experiments on CFD and CSD benchmarks comparing to baselines including MGN, MGN Tree, MARIO, MMGP, ViT, and FNO are performed, showing erors can be significantly reduced.

**Compliance With Llm Reviewing Policy:**

Affirmed.

**Final Justification:**

The authors have provided detailed rebuttal information for all reviewers. In general I think the core design of this work is reasonable. I will keep my already positive rating.

**Key Questions For Authors:**

- What is the computational cost of the tree construction, especially when the entire graph contains millions to billions of nodes?
- Is there any reason why the extra edges are updated every T epochs? Is there any guidance how T should be chosen?

**Limitations:**

Yes, limitations have been discussed in the Conclusion and the Impact Statement sections.

**Strengths And Weaknesses:**

Strengths:
- Using error information to perform graph augmentation is reasonable and promising.

Weaknesses:
- The claim that "MiSe-GNN is the first to leverage a learned a posteriori error indicator to drive a discrete, dynamic graph augmentation strategy in computational mechanics" may be inappropriate, because as early as MeshGraphNet, error estimation has been used for remeshing, which surely also leads to dynamic graph adaptation.
- The test problems seem relatively simple and small in scale. It will be interesting to see the effectiveness and efficiency of proposed mechanism in larger-scale, 3D problems.

---

> ### Author Rebuttal · Authors · 2026-03-31
>
> We sincerely thank the reviewer for the positive feedback and for highlighting both the promise and the current limits of the approach. We appreciate these comments and will revise the manuscript to clarify the novelty claim and provide additional details on scalability and hyperparameter choices.
>
> You are correct that the current novelty claim is too broad. We would like to clarify that our claim is not to be the first to use error information in computational mechanics, as error estimation has long been used for remeshing and adaptive refinement.
>
> Our more precise contribution is using a learned a posteriori error signal to drive discrete graph connectivity augmentation on a fixed mesh graph, rather than modifying the physical mesh itself. In other words, MiSe-GNN adapts message-passing connectivity while preserving the underlying discretization. We will revise the wording in the introduction accordingly to make this distinction clearer.
>
> We also agree that scaling to larger and especially 3D problems is an important next step. We already note this as a limitation: extending MiSe-GNN to very large or 3D meshes will require more scalable dynamic connectivity and tree construction. We will emphasize this limitation more clearly in the revision. And we are currently developing models to address these limitations.
>
> Q1: **Tree construction cost and scale**
>
> The table below details the average time required to build error edges per sample across the evaluated datasets:
>
> | Dataset | Number of Samples | Avg. Time to Build Error Edges / Sample |
> | :--- | :---: | :---: |
> | 2D MultiScHypEl | 1140 | 0.435(s) |
> | 2D profile      | 400  | 0.519(s) |
> | VKI-LS59        | 839  | 0.579(s) |
> | Tensile2d       | 700  | 0.167(s) |
>
> We included an accuracy–time budget study on 2D profile. Although MiSe-GNN has a higher per-epoch cost, it reaches MGN’s $100 / 200 / 500$ epoch accuracy in $15.08 / 16.68 / 101.84$ minutes, versus $44.74 / 69.86 / 177.90$ for MGN. Thus, the added cost is amortized by faster convergence and better anytime performance.
>
> Q2: **Why update every T epochs, and how should T be chosen?**
>
> Our adaptive tree algorithm relies on the model's error predictions to construct augmented edges. Because these predictions continously improve during training, the graph connectivity must be dynamically updated to reflect the lastest error map. But, to avoid the computational cost of rebuilding the graph at every epoch, we perform this update periodically every $T$ epochs.
>
> We conducted an ablation over $T$ in Appendix E.2, and the results demonstrate the expected trade-off: more frequent updates improve accuracy but increase augmentation overhead, while less frequent updates reduce this overhead at the cost of slight accuracy. We selected $T=10$ as a practical balance between these two effects, and we will clarify this rationale more explicitly in the main text of the revised paper.

---

> > ### Author Rebuttal · Reviewer_C3UB · 2026-04-01
> >
> > My concerns regarding claims, experimental settings, computational cost, and edge update frequency, have been responded in the rebuttal, within which some have been resolved. Reviewer sVaz has proposed some very good points, which also have been detailedly addressed. Hence overall I consider the rebuttal sufficient.

---

### Official Review · Reviewer_nmMX · 2026-03-10

**Soundness:** 4
**Presentation:** 3
**Significance:** 3
**Originality:** 4
**Overall Recommendation:** 5
**Confidence:** 4

**Summary:**

This paper introduces MiSe-GNN, a graph augmentation strategy driven by error prediction for GNN-based PDE surrogates.
The GNN has two output heads that predict a field and the corresponding error relative to the ground truth, respectively.
The authors introduce two graphs: the raw graph and the augmented one.
At each training time iteration, a first forward pass with the raw graph is used to compute the true error and the MSE relative to the predicted error.
Then, the field predictions of a second forward pass with the augmented graph are used to compute the MSE relative to the groundtruth field.
The final loss is the sum of the two MSE values.
During training, the augmented graphs are updated periodically based on predicted errors. This algorithm recursively divides the nodes along the axis with the maximum error variance at the location of the highest error node (pivot) so as to build an error-based tree representation of the nodes. The pivots are then connected to their k-hop ancestors if their error exceeds a threshold. The new connections are accumulated with those of the raw graph.
During inference, a first forward pass provides the error estimations that are used to create the augmented graph and a second forward pass returns the field predictions.

**Compliance With Llm Reviewing Policy:**

Affirmed.

**Key Questions For Authors:**

1. Could you provide the learning rate and learning rate scheduler used for reproducibility?
2. It is not specified which information is backpropagated in the network. Should we assume that no gradient stopping has been applied? Is $\epsilon^*$ detached from the backpropagation graph? Is there a gradient stopping before the error prediction head?
3. Have you tested training two different GNNs to make predictions with the raw graphs and the augmented graphs, respectively? I am not asking you to conduct these experiments, as I do not believe they are necessary for your article. I am just interested in the outcome.
4. Do the authors plan to publish their code?

**Limitations:**

yes

**Strengths And Weaknesses:**

Strengths:
1. The authors propose an original approach to automatically increasing graph connectivity where it matters the most.
2. They have conducted extensive experiments to rigorously demonstrate that the proposed graph augmentation consistently improves training efficiency across various tasks.
3. Several geometric encodings were also studied to show the complementarity between error-driven graph augmentation and geometric encoding.
4. The paper is well-structured, well-written and well-illustrated.

Weaknesses:
1. Since AUGUR is closed-source, comparisons with recent and open-source transformer-based approaches such as Transolver, Erwin, (Anchor-Based) Universal Physics Transformer, Geometry Aware Operator Transformer (GAOT), etc., are lacking.
2. The fact that two inferences are required may lead to redundant computation. It could have been interesting to partially reuse the features of the first forward pass with the raw graph when performing the forward pass for better efficiency.
3. There are missing implementation details, including the learning rate and learning rate scheduler.

Minor weakness: the legends of Figures 3 and 4 could be more detailed.

---

> ### Author Rebuttal · Authors · 2026-03-31
>
> We sincerely thank the reviewer for the positive evaluation and for the constructive questions. We are glad that the reviewer found the idea of error-guided graph augmentation interesting and the experimental evaluation convincing.
>
> We compared our model to the public baslines of the PLAID benchmark, which included various model types such as INR (MARIO), Transformer (Augur & Vi-Transformer), MMGP, FNO, etc. We agree that reusing intermediate features from the first pass is a promising efficiency direction and will mention it as future work.
>
> We respond to the specific questions below and will incorporate the requested implementation details in the revised version.
>
> Q1: **Learning rate and scheduler**
>
> We used Adam with learning rate $1e-3$ and no scheduler. We will add the exact optimization settings in the revision.
>
> Q2: **Gradient flow, detach question**
>
> Yes, a stop gradient operation is applied. We agree that this is a important implementation detail currently omitted from Eq. (4), and we will make it explicit in the revised paper.
>
> The training process for the error head works as follows:
>
> - Target construction: the ground truth error target $\epsilon^*$ is computed based on the field prediction from the base graph ($\hat{y}_{base}$).
>
> - Stop gradient (detach): once computed, we apply a .detach() operation to $\epsilon^*$. This treats the computed error as a strictly fixed target.
>
> - Gradient flow: during backpropagation, the gradients from the error loss $L_{err}$ only flow backward through the predicted error $\hat{\epsilon}\_{base}$ to update the error head and the shared encoder-processor. Absolutely no gradients from $L_{err}$ flow backward through the target $\epsilon^*$ to alter the field prediction $\hat{y}_{base}$ or the field head.
>
> This design ensures that the error head learns to predict errors without detrimentally forcing the field head to change its predictions merely to minimize the error loss.
>
> Q3: **Two separate GNNs for raw and augmented graphs**
>
> We tested a two-model pipeline in which one error model predicts error on the base graph, and another predicts the field on the augmeneted graph from that predicted error.
>
> `Base graph -> Error model -> Error prediction -> Augmented graph -> Field model -> Field prediction`
>
> On the **u1** task of **2D MultiScHypEl**, this two-model variant improves over MGN and MGN Tree, but remains below the shared dual-head MiSe-GNN:
>
> | Model | Test RRMSE |
> |---|---:|
> | MGN | 0.0419 |
> | MGN Tree | 0.0249 |
> | Multi-model GNN | 0.0221 |
> | MiSe-GNN | **0.0185** |
>
> This suggests that the gain is not only from graph refinement itself, but also from joint representation learning: sharing the encoder-processor between field prediction and error prediction yields a stronger error signal and better final accuracy than decoupling the two tasks into separate models.
>
> Q4: **Code release**
>
> We plan to release the code after the review process, subject to anonymity constraints. Please note that the codebase is already fully prepared for public release, it has been thoroughly cleaned, documented, and well-organized.

---

> > ### Author Rebuttal · Reviewer_nmMX · 2026-04-02
> >
> > My concerns about missing implementation details, as well as my request for an additional experiment, have been fully addressed. I am satisfied with this rebuttal.

---

### Official Review · Reviewer_sVaz · 2026-03-13

**Soundness:** 3
**Presentation:** 3
**Significance:** 3
**Originality:** 3
**Overall Recommendation:** 5
**Confidence:** 4

**Summary:**

The paper studies graph augmentation for mesh-based steady-state PDE surrogates and proposes MiSe-GNN, a dual-head GNN that predicts both the field and a node-wise error signal, then uses the predicted error to add long-range edges through an adaptive tree. The idea is interesting and the empirical results are promising across several CFD/CSD datasets.

**Compliance With Llm Reviewing Policy:**

Affirmed.

**Final Justification:**

I find the presented rebuttal along with detail analysis-backed claims satisfactory for my concerns.

**Key Questions For Authors:**

1. Since the error target is defined from the prediction on the base graph, while the final field prediction is made on the augmented graph (Eq (4)), why is this the right supervision signal for learning the rewiring policy? Did you consider alternatives based on augmented-graph residuals or other alignment strategies?
2. For shock-containing or otherwise strongly anisotropic steady flows, how do you justify that the proposed undirected tree/ancestor edges are physically appropriate, rather than introducing shortcuts that are not aligned with the underlying transport or characteristic structure (For reference – in a steady compressible flow around an airfoil, that means the added edges are not constrained by characteristics, streamline direction, or shock normal direction)?
3. Can you compare MiSe-GNN against simpler equal-budget augmentation baselines (e.g., random long-range edges, distance-based edges, or static hierarchical edges) to isolate whether the benefit comes from error guidance specifically?
4. The proposed method seems particularly motivated by regimes where neighboring nodes may not have smoothly varying targets (e.g., shocks or sharp gradients). Did the authors analyze whether the gains of MiSe-GNN correlate with local homophily/heterophily of the target field, or compare against methods designed specifically for low-homophily message passing?

**Limitations:**

The paper would benefit from a short discussion of when long-range graph augmentation is physically appropriate in steady CFD/CSD, and when it may conflict with strongly directional or discontinuous phenomena.

**Strengths And Weaknesses:**

Strengths
1. The paper addresses an important and practically relevant problem: improving long-range information propagation in mesh-based PDE surrogates.
2. The proposed error-guided augmentation mechanism is intuitive and reasonably interpretable, and the visualization of where extra edges are added is a useful aspect of the work.
3. The empirical evaluation is broad and the reported gains over strong baselines, along with efficiency and hyper-parameter studies, make the paper technically substantial.

Weaknesses
1. The error head is supervised using residuals on the base graph, while the final prediction is made on the augmented graph (Eq (4)), creating a potential mismatch between the auxiliary supervision signal and the actual prediction setting.
2. The physical validity of the added long-range edges is not fully convincing for problems with strong anisotropy or discontinuities (e.g., shocks), since the augmentation is based on undirected, geometry/error-driven tree connections rather than physics-aware directional structure.
3. It remains unclear whether the gains come specifically from the proposed error-driven tree rewiring, as opposed to simply adding useful extra long-range edges; the paper would be stronger with simpler equal-budget augmentation controls.

---

> ### Author Rebuttal · Authors · 2026-03-31
>
> We sincerely thank the thoughtful and technically detailed feedback. We appreciate the opportunity to clarify several design choices and provide additional controls that further support the proposed error-guided augmentation mechanism.
>
> Q1: **Supervision on the base graph**
>
> This separation is intentional rather than a mismatch. The error head is trained to diagnose where the base graph fails before intervention. If the error supervision target were defined on the changing augmented graph, successful augmentation would partly erase the very signal used to decide where rewiring is needed, causing the controller target to move during training.
>
> We also tested the alternative directly: an iterative variant that predicts the error on the current augmented graph and new edges are rebuilt from that prediction. While this variant trains faster, it consistently produces less accurate solutions. For example:
>
> 2D MultiScHypEl (u1 RRMSE):
> MiSe-GNN: 0.0185
> Iterative variant: 0.0234
>
> 2D Profile (Pressure RRMSE):
> MiSe-GNN: 0.0290
> Iterative variant: 0.0333
>
> These results support the design choice of learning a stable pre-intervention diagnostic signal on the base graph $G_b$.
>
> Q2: **Undirected edges for anisotropic flows**
>
> We agree that the added edges are not characteristic-aware, and we do not present them as a physics-exact transport graph. Instead, they act as information pathways for message passing, allowing the network to exchange information across regions where purely local connectivity is insufficient.
>
> Directional information is still available to the model through edge attributes such as relative displacement and optional geometric encodings. Therefore, while the graph topology itself is undirected, the message functions can still learn anisotropic responses from these features.
>
> We therefore view the current approach as selective multiscale communication, rather than a physics-exact discretization. Incorporating physics-aware directional rewiring is an interesting extension, and we will clarify this limitation more explicitly in the revised manuscript.
>
> Q3: **Equal-budget simpler baselines**
>
> To isolate whether gains stem specifically from error-guidance, we evaluated two equal-budget controls: Random edges and Distance-based edges, strictly matching MiSe-GNN’s average extra-edge count over training.
>
> While distance-based augmentation can be useful, it is highly sensitive to mesh density artifacts. For instance, it performs poorly on Tensile2d (where denser boundary sampling misleadingly attracts edges) and VKI-LS59. To separate the augmentation policy from mesh density cues, we also tested a Uniform Mesh version of 2D Profile. In this unbiased setting, purely geometry-driven heuristics lose their signal, whereas MiSe-GNN remains highly effective because it explicitly targets predictive errors.
>
> As summarized in the table below (15 MP / hidden size 80), MiSe-GNN consistently outperforms all equal-budget controls. This confirms that the performance leap is driven by how the extra-edge budget is allocated (error-guided routing), not merely by adding more edges.
>
> **Overall Summary (Macro RRMSE & MiSe-GNN Improvement %)**
>
> | Model | Extra-edge policy | 2D Profile (Base) | 2D Profile (Uniform) | 2D MultiScHypEl | VKI-LS59 | Tensile2d |
> | :--- | :--- | ---: | ---: | ---: | ---: | ---: |
> | MGN | none | 0.0356 (+17.0%) | 0.0274 (+24.5%) | 0.0269 (+16.6%) | 0.0310 (+49.7%) | 0.0282 (+89.2%) |
> | MGN Tree | static hierarchical | 0.0376 (+21.4%) | 0.0250 (+17.3%) | 0.0261 (+14.1%) | 0.0191 (+18.3%) | 0.0039 (+21.6%) |
> | MGN Random | random | 0.0383 (+22.9%) | 0.0251 (+17.4%) | 0.0242 (+7.4%) | 0.0244 (+36.1%) | 0.0039 (+21.5%) |
> | MGN Dist-based | distance-based | 0.0368 (+19.7%) | 0.0266 (+22.3%) | 0.0251 (+10.9%) | 0.0307 (+49.3%) | 0.0256 (+88.1%) |
> | **MiSe-GNN** | **error-guided** | **0.0295** | **0.0207** | **0.0224** | **0.0156** | **0.0031** |
>
> Q4: **Homophily/Heterophily**
>
> This is a very interesting perspective. We thank the reviewer for the insightful connection between physical discontinuities (e.g., shocks) to low homophily. Standard local message passing struggles in these highly dissimilar regions. MiSe-GNN inherently circumvents this bottleneck: qualitatively, the predicted error identifies where local propagation fails, routing adaptive edges directly to these low-homophily areas (Fig. 7, App. Fig. 27). Quantitatively, our Top-5% Error Mass Capture (App. G, Table 28) confirms this targeting capability, reaching $0.90$ on shock-dominated fields. We will incorporate this valuable perspective into the revision and propose formal homophily analysis as future work.
>
> Due to rebuttal space, we provide the full equal-budget tables and visualizations: https://imgur.com/a/Bx6eeTp
>
> Consistent with a posteriori error-based refinement, MiSe-GNN allocates extra connectivity near shock/high-error regions, whereas distance-based edges are driven mainly by geometry/mesh density and can be less physically appropriate.

---

> > ### Author Rebuttal · Reviewer_sVaz · 2026-04-02
> >
> > I thank the authors for proving clarifications to my concern. I find them satisfactory.
> > I appreciate the detailed analysis presented in the rebuttal and addressing my point on homophily as the possible connection between physical fields and refinement.
> > I will update my scores and good luck!

---

### Official Review · Reviewer_eBHr · 2026-03-19

**Soundness:** 3
**Presentation:** 2
**Significance:** 3
**Originality:** 2
**Overall Recommendation:** 4
**Confidence:** 3

**Summary:**

The proposed method, Mise-GNN, is an extension of the GNN family of surrogate PDE solvers. It includes an additional error-head that captures the region of large errors. Then, it can adaptively augment the graph connectivity to allocate more capacity to the harder region with complex physics (eg, shocks, wakes). The method is compared with prior mesh graph nets again, and other surrogate models like neural operators, primarily for solving CFD problems. The results consistently show that the prediction and dynamically adaptive tree algorithm reduces the overall error of the surrogate model's predictions.

**Compliance With Llm Reviewing Policy:**

Affirmed.

**Final Justification:**

My main concerns were about the writing and presentation, and your rebuttal clarified most of these issues. I ask the authors to carefully address these points in the revision and update my score.

**Key Questions For Authors:**

1.	The proposed model periodically updates error-guided edges to improve long-range information propagation. Could you also remove edges where the error is lower, for efficiency?
2.	A higher error at a predicted node always implies that a long-range connection is required. It might be that they simply need more connections locally. In this case, I am not convinced by the pitch of the paper, which focuses heavily on long-range connections.
3.	In Fig. 1, does the same point get chosen again and again?

**Limitations:**

yes

**Strengths And Weaknesses:**

Soundness:

++ The proposed idea of learning to predict error and refining the graph to focus on the higher-error regions is quite sound. The evaluations sufficiently support this claim.

Presentation:

-- Although the method is generally well-written, there are sections that are not, and it severely restricts the understanding and ease of reading. For instance, the proposed method learns to predict errors during training and adaptively refines the graph structure _during inference_.  This was not clear until I read the last paragraph of Sec. 2.3 and 3.2.3, as it was not described clearly earlier. The current version _when_ the error is predicted, and _when_ it is used. It needs to be mentioned upfront in the introduction and the method overview!
Another instance is Sec. 2.1. It should be significantly improved in terms of writing (you could reorganize what goes in the appendix and the main paper). Edge construction algorithm is unclear, especially what different k-hops imply, what the recommended etc.

Significance:

++ Solving fluid and structural dynamics, especially using neural methods, is highly significant and has a broad impact.

Originality:

++ The work is relatively original, although many variations of adaptive graph networks (eg, w/ remeshing or hierarchical message passing) have already been explored in the past.

---

> ### Author Rebuttal · Authors · 2026-03-31
>
> We sincerely thank the reviewer for the constructive feedback and for the helpful clarity questions.
>
> We agree that the training/inference schedule could be explained earlier in the paper and will revise the introduction and method overview accordingly.
>
> Specifically, we will explicitly detail the process as follows: During training, the error head always predicts the error map $\hat{\epsilon}\_{base}$ on the fixed base graph $G_b$. Every $T$ epochs, the augmented graph $G_a=(V,E_b\cup E^\star)$ is rebuilt from this predicted error, and the field head then predicts the physical fields on $G_a$ (Sec. 2.3, Eqs. (2–4), Fig. 4). During inference, the same two-stage procedure is applied: first predict $\hat{\epsilon}_{base}$ on $G_b$, then construct $E^\star$, and finally predict the field on $G_a$.
>
> The decoupling of the physical field prediction and the error prediction onto two distinct graphs is intentional. The error head diagnoses where the base graph fails, while the field head exploits the repaired connectivity. Importantly, augmented edges are not accumulated over time. They are refreshed from the current error map, so regions that become well-predicted stop receiving additional connectivity. The original mesh edges are always retained and the augmentation only complements them.
>
>
> Q1: **Could low error edges be removed ?**
>
> Thank you for the question, indeed, yes the augmentation is refreshed rather than accumulated. At each update, nodes that no longer exhibit high predicted error not only stop receiving new edges, but their previously assigned extra edges are also pruned away. We indeed observe this behavior during training. For example, when tracking one sample from 2D Profile, the number of added edges decreases as the model improves:
>
> **Number of added edges during training**
>
> | Epoch | Number of extra-edges |
> |---|---:|
> | 10   | 72692 |
> | 500  | 33567 |
> | 1000 | 17484 |
>
> We intentionally keep the original mesh edges because they encode the local discretization couplings of the underlying PDE. Removing them degrades performance: for the u1 task of 2D MultiScHypEl, MiSe-GNN achieves $0.0185$ RRMSE with the original mesh retained versus $0.0243$ when the mesh is replaced by only the additional edges introduced by the MGN Tree baseline.
>
> On the **u1** task of **2D MultiScHypEl**:
>
> | Model | Test RRMSE |
> |---|---:|
> | MiSe-GNN (tree-only base graph, original mesh removed) | 0.0243 |
> | MiSe-GNN (full base graph, original mesh retained)| **0.0185** |
>
> Thus the design preserves reliable local physics while adaptively adding connections only where the model predicts they are needed. Such details will and should be inserted as suggested by the reviewer in the revised version.
>
> Q2: **Does higher error always imply that long-range connectivity is required ?**
>
> On whether higher error necessarily means long-range edges: our claim is more modest. Instead, predicted error is only a signal that the fixed base graph may be insufficient in that region. The adaptive tree then adds multiscale connectivity in those regions. In practice this mechanism increases connectivity at multiple scales. When high-error regions are spatially concentrated, recursive partitioning places pivots more densely, which naturally increases local and mid-range connectivity around those difficult regions. Therefore the method should be viewed as adaptive multiscale connectivity, rather than purely long-range augmentation.
>
> Q3: **Can the same point be chosen repeatedly in Fig. 1 ?**
>
> On Fig. 1: pivots are chosen within the current subdomain. The same node is not repeatedly selected within one leaf, across descendants it can only reappear if it is still highest-error valid splitter in that smaller subdomain. When the highest error candidate would create an empty partition, the algorithm retries with the next candidate, this is exactly why Step 5 uses the second-highest error point.

---

> > ### Author Rebuttal · Reviewer_eBHr · 2026-04-03
> >
> > Thanks for the rebuttal! My main concerns were the writing and presentation, and your rebuttal clarified most of them. I ask the authors to carefully address these points in the revision, and I am updating my score.

---

### Decision · Program_Chairs · 2026-04-30

**Decision:**

Accept (regular)

**Comment:**

This paper received a unanimously positive recommendation from all reviewers after the post-rebuttal discussion. Reviewers in general appreciate the paper's novelty and technical contribution. Several reviewers initially questioned the paper's technical exposition (e.g., clarifying some claims and technical details), the source of its performance gains, its comparisons with open-source baselines, and its extensions to large-scale problems. The rebuttal and the reviewer-author discussion effectively addressed these concerns. Therefore, I recommend acceptance.